# TMEM95 is a sperm membrane protein essential for mammalian fertilization

Ismael Lamas-Toranzo[1†], Julieta G Hamze[2†], Enrica Bianchi[3], Beatriz Fernández-Fuertes[4,5], Serafín Pérez-Cerezales[1], Ricardo Laguna-Barraza[1], Raúl Fernández-González[1], Pat Lonergan[4], Alfonso Gutiérrez-Adán[1], Gavin J Wright[3], María Jiménez-Movilla[2*], Pablo Bermejo-Álvarez[1*]

[1]Animal Reproduction Department, INIA, Madrid, Spain; [2]Department of Cell Biology and Histology, Medical School, University of Murcia, IMIB-Arrixaca, Murcia, Spain; [3]Cell Surface Signalling Laboratory, Wellcome Trust Sanger Institute, Cambridge, United Kingdom; [4]School of Agriculture and Food Science, University College Dublin, Dublin, Ireland; [5]Department of Biology, Faculty of Sciences, Institute of Food and Agricultural Technology, University of Girona, Girona, Spain

**Abstract** The fusion of gamete membranes during fertilization is an essential process for sexual reproduction. Despite its importance, only three proteins are known to be indispensable for sperm-egg membrane fusion: the sperm proteins IZUMO1 and SPACA6, and the egg protein JUNO. Here we demonstrate that another sperm protein, TMEM95, is necessary for sperm-egg interaction. TMEM95 ablation in mice caused complete male-specific infertility. Sperm lacking this protein were morphologically normal exhibited normal motility, and could penetrate the zona pellucida and bind to the oolemma. However, once bound to the oolemma, TMEM95-deficient sperm were unable to fuse with the egg membrane or penetrate into the ooplasm, and fertilization could only be achieved by mechanical injection of one sperm into the ooplasm, thereby bypassing membrane fusion. These data demonstrate that TMEM95 is essential for mammalian fertilization.

*For correspondence:
mariajm@um.es (MíJé-M);
bermejo.pablo@inia.es (PB-Á)

†These authors contributed
equally to this work

Competing interests: The
authors declare that no
competing interests exist.

Reviewing editor: Polina V
Lishko, University of California,
Berkeley, United States

## Introduction

In sexually reproducing species, life begins with the fusion of two gametes during fertilization. Mammalian fertilization requires the sperm to pass through a glycoprotein coat that surrounds the egg, termed the zona pellucida, and once in the perivitelline space it must fuse its membrane with that of the egg to deliver the paternal genetic material into the ooplasm. Despite its importance in sexual reproduction, the molecular mechanisms behind the fusion of gamete membranes remain largely unknown. Currently, only three proteins have been demonstrated to be essential for sperm-egg membrane interaction: the sperm proteins IZUMO1 (*Inoue et al., 2005*) and SPACA6 (*Lorenzetti et al., 2014*), and the egg protein JUNO (*Bianchi et al., 2014*). The ablation of another egg protein, CD9 (*Kaji et al., 2000*; *Le Naour et al., 2000*; *Miyado et al., 2000*), also leads to a significant impairment in sperm-egg membrane fusion, but its ablation does not cause complete infertility. Similarly, very recently, the ablation of the sperm protein FIMP has been reported to cause severe subfertility due to gamete fusion defects (*Fujihara et al., 2020*).

IZUMO1 was the first protein to be shown to be critical for gamete fusion (*Inoue et al., 2005*). Sperm lacking IZUMO1 exhibit normal morphology and motility and are able to pass through the zona pellucida, but incapable of fusing their membrane with that of the egg. The search for the egg binding partner of IZUMO1 lasted almost a decade, until the protein JUNO was shown to be required for gamete membrane binding through direct interaction with IZUMO1 (*Bianchi et al., 2014*). Here, we demonstrate by gene editing in mice that besides these two proteins, another sperm protein, TMEM95, plays an essential role in fertilization.

## Results

### TTMEM95 is a sperm protein dispensable for embryo development that relocalizes to the equatorial region after acrosomal reaction

*TMEM95* coding region was located in a 1386 kb segment of extended homozygosity in 40 sub-fertile bulls identified in a Genome Wide Association Study carried out on the Fleckvieh breed population (*Pausch et al., 2014*). Within this segment, containing 80 transcripts, a non-sense mutation in TMEM95 coding region that reduces protein length to 151 amino acids was identified (*Pausch et al., 2014*). The semen of bulls carrying two of these partially truncated alleles exhibited normal motility parameters but a dramatic reduction in egg penetration was evident following in vitro fertilization (*Fernandez-Fuertes et al., 2017*). In silico protein folding analysis shows that TMEM95 protein contains a single transmembrane domain and a secondary structure formed by β-hairpin and α-helix which is remarkably similar to that found on IZUMO1 protein and termed 'IZUMO1 domain' (*Ellerman et al., 2009*), thereby suggesting a possible role in sperm-egg fusion (*Figure 1A*). Similar bioinformatics predictions have been reported for bovine IZUMO1 (*Zhang et al., 2016*).

To elucidate the role of TMEM95 in fertilization, we generated *Tmem95*-deficient mice by CRISPR-directed mutagenesis upon microinjection of mouse embryos (CBAXC57BL6 F1 hybrids) at the zygote stage (*Wang et al., 2013*). sgRNA was designed against the first exon of *Tmem95* using bioinformatic tools to minimize the chances of off-target genome editing (https://crispr.mit.edu). Genome editing was confirmed in the pups derived from CRISPR-injected embryos by clonal sequencing, and one edited founder female harbouring a frame-disrupting allele and an in-frame indel was selected to obtain heterozygous offspring by crossing with wild-type males of the same genetic background. Frame-disrupting indel (knock-out allele) consisted of a 4 bp substitution and a 10 bp deletion (*Figure 1B*) resulting in a predicted truncated peptide of 53 amino acids sharing a region of only 33 amino acids homologous to the N-terminus of TMEM95 protein (*Figure 1C*). The absence of off-target mutations in the founder female was confirmed by sequencing the five most probable off-target sites (*Figure 1—figure supplement 1*). F1 heterozygous pups showed similar frequencies for both in-frame and frame-disrupting indels (9:11, respectively) and those pups harbouring the frame-disrupting allele (KO allele) were selected and intercrossed to obtain an F2 generation composed of *Tmem95$^{+/+}$* (wild-type, WT), *Tmem95$^{+/-}$* (heterozygous, Hz) and *Tmem95$^{-/-}$* (knock-out, KO) pups. The intercross of the heterozygous F1 generation resulted in normal litter sizes, and alleles segregated in a Mendelian distribution (20:43:26 for WT:Hz:KO), indicating that *Tmem95* haploinsufficiency did not impair reproductive performance and that *Tmem95* bi-allelic ablation did not result in embryonic developmental failure.

Further confirmation of gene ablation was achieved by Western blotting in 3 samples of WT and KO sperm. A ~ 20 kDa band detected in WT samples was absent in KO samples (*Figure 1D*, *Figure 1—figure supplement 2A and C–D*). The size of the band was compatible with TMEM95 predicted size and peptide sequencing identified TMEM95 protein on that band (*Table 1*). IZUMO1 protein was expressed exclusively in testis and sperm (*Figure 1—figure supplement 2B–C*) and at similar levels in WT and KO sperm (*Figure 1D*, *Figure 1—figure supplement 2E*). Immunocytochemistry also demonstrated the absence of TMEM95 protein in sperm obtained from KO males (*Figure 1E*). In WT acrosome-intact sperm, TMEM95 protein was present in the acrosomal cap, in agreement with previous findings in cattle (*Fernandez-Fuertes et al., 2017*; *Pausch et al., 2014*). The specific localization of TMEM95 within the acrosomal cap could not be determined as paraformaldehyde (PFA) fixed sperm untreated with Triton-X100 displayed signals for the inner acrosomal markers PNA and IZUMO1, suggesting that acrosomal membrane was permeable to antibodies (*Figure 1—figure supplement 3A*). Following acrosome reaction, TMEM95 relocalized to the equatorial segment (*Figure 1F*), the place where gamete membrane fusion occurs. To test the possibility that TMEM95 ablation may impair IZUMO1 translocation, we determined IZUMO1 translocation in acrosome-reacted WT and KO sperm, observing that IZUMO1 translocation also occurred in the absence of TMEM95 (*Figure 1—figure supplement 3B*).

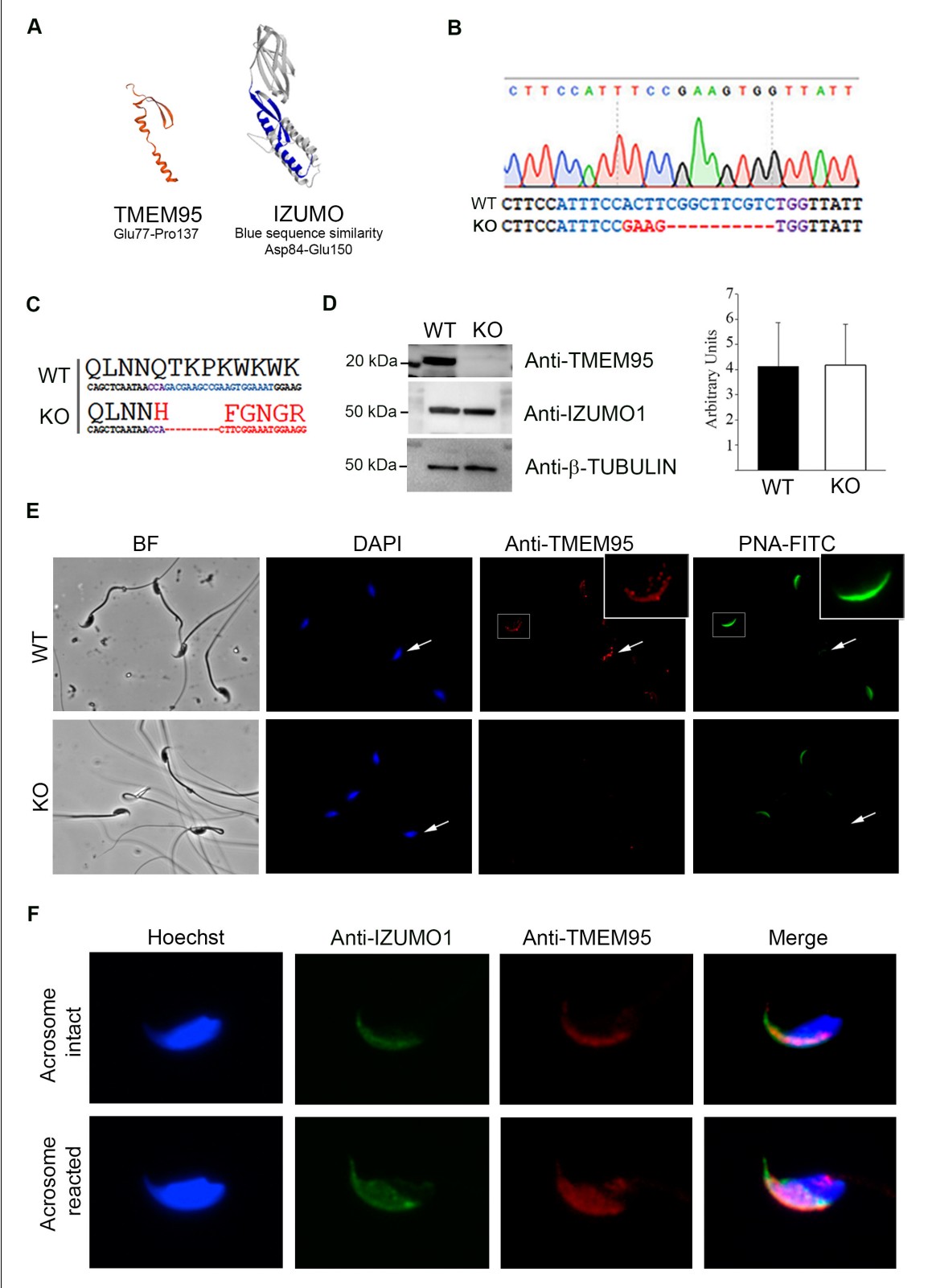

**Figure 1.** Generation of TMEM95 deficient mice. (**A**) Structure prediction of TMEM95 protein (left) using IZUMO1 (right) as template, created by SWISS-MODEL software. (**B**) *Tmem95* KO allele generated following CRISPR-mediated edition. CRISPR target sequence and PAM are depicted in blue and purple letters, respectively. (**C**) The deletion of 10 bp altered *Tmem95* ORF. Large letters indicate the aminoacid sequence corresponding to the codons (DNA sequence) shown in smaller letters below. (**D**) Western Blot images for TMEM95, IZUMO1 and β-tubulin proteins from protein extracts

*Figure 1 continued on next page*

*Figure 1 continued*

from WT or KO sperm. Graph on right indicates the abundance of IZUMO1 in WT and KO extracts. (**E**) Immunocytochemistry images of KO and WT sperm stained with an antibody against TMEM95 and the acrosomal stain PNA. TMEM95 localized to the acrosomal cap in acrosome intact sperm and in the equatorial segment after acrosome reaction. (**F**) Immunocytochemistry images of acrosome intact (upper images) or reacted (lower images) WT sperm stained against IZUMO1 and TMEM95. Both proteins relocalize to the equatorial segment following acrosome reaction.

The online version of this article includes the following figure supplement(s) for figure 1:

**Figure supplement 1.** Offtarget analysis.
**Figure supplement 2.** WB images related to *Figure 1D*.
**Figure supplement 3.** Acrosomal cap localization of TMEM95 and IZUMO1 and IZUMO1 relocalization in WT and KO sperm.

## *Tmem95*-deficient sperm cannot fuse with eggs

*Tmem95* mutant males were healthy, grossly normal and exhibited normal mating behaviour evidenced by the presence of copulatory plugs in females. However, no offspring were obtained from the cross of KO males and fertile WT females after the observation of 24 copulatory plugs from 8 pairs. In contrast, Hz males exhibited normal fertility, comparable to WT counterparts, indicating that haploinsufficiency had no effect on reproductive performance. To determine whether ejaculated sperm from *Tmem95*-null males were able to fertilize in vivo, we mated 4 WT, Hz or KO males with superovulated WT females and recovered the embryos 1.5 days after mating by oviductal flushing. Over 90% of the embryos recovered from females mated with WT or Hz males were cleaved at the time of recovery, whereas no divided embryos were recovered from females mated with KO males (*Figure 2A*). Similar results were obtained after in vitro fertilization (IVF); sperm recovered from KO males were apparently unable to fertilize cumulus-oocyte complexes as no embryonic cleavage occurred. By contrast, sperm recovered from Hz males yielded similar rates of embryo cleavage than those obtained from WT counterparts (~70%, *Figure 2B*). Following in vivo and in vitro fertilization with sperm lacking TMEM95, uncleaved eggs displayed a similar appearance to those non-fertilized by IZUMO1-deficient sperm (*Inoue et al., 2005*): the perivitelline space contained multiple sperm, probably due to the absence of sperm penetration into the egg and associated zona hardening, allowing multiple sperm to travel across the zona pellucida (*Figure 2C*, *Video 1*). TMEM95-null sperm were in close contact with the oolemma but were still unable to penetrate into the ooplasm. This result indicates that TMEM95-deficient sperm are able to pass through the zona pellucida and exhibit an overtly normal motility. To confirm that TMEM95 ablation did not impair sperm motility, we performed computer aided sperm analysis (CASA), observing no differences in any of the kinetics parameters analysed (*Figure 2D*). TMEM95-deficient sperm were indistinguishable from WT sperm by contrast microscopy (*Figure 1E*), and the normal morphology of TMEM95-deficient sperm was also assessed by Transmission Electron Microscopy, which showed no detectable abnormalities at the submicroscopic level (*Figure 2E*).

The absence of zona hardening strongly suggested a failure in sperm penetration into the egg. To further test this failure, unfertilized eggs co-incubated with *Tmem95*-deficient sperm were stained with the DNA-binding stain DAPI. These samples were also stained with the acrosome staining PNA to test whether the TMEM95-null sperm that had penetrated the zona pellucida had successfully undergone the acrosome reaction, an essential step prior to membrane fusion (*Satouh et al., 2012*). No sperm heads or male pronuclei were detected in the ooplasm of eggs co-incubated with sperm lacking TMEM95, and those sperm accumulated in the perivitelline space were PNA negative (*Figure 2—figure supplement 1A*). These results confirm that TMEM95-null sperm had undergone the acrosome reaction but failed to fuse with the egg membrane. Subsequently, we tested whether

**Table 1.** Details of peptides identified by MS/MS from 20 kDa gel band.

| Protein name | Score | Spi % | Sequence |
|---|---|---|---|
| TMEM95 | 5.33 | 68.8 | LLLCIFGIVLLsGVVSLQ |
| TMEM95 | 5.17 | 59.8 | LLLCIFGIVLLsGVVSLQ |
| TMEM95 | 4.12 | 46.1 | LLSGVVSLQVEY |
| TMEM95 | 3.42 | 53 | KTRYP |

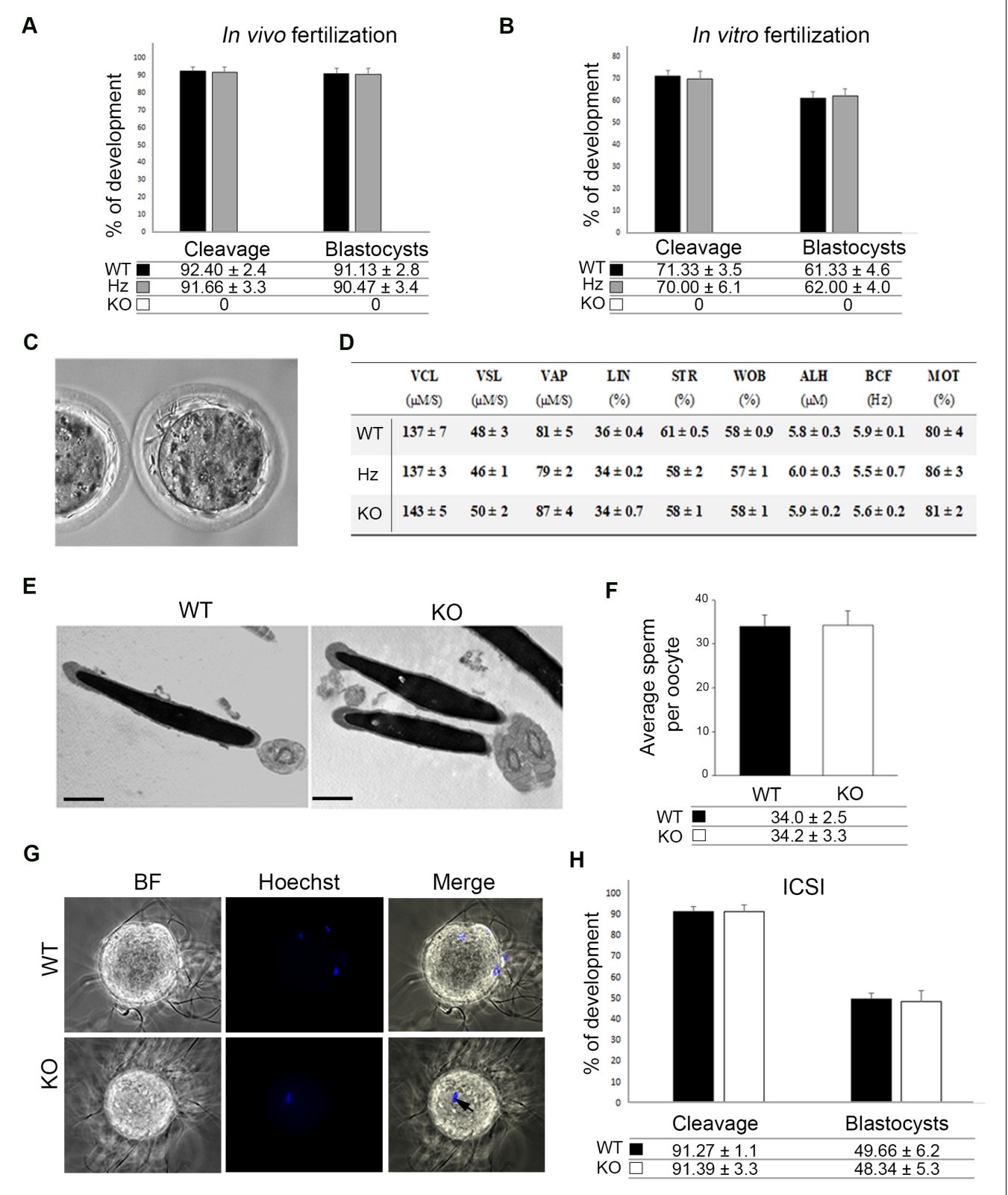

**Figure 2.** Reproductive performance of TMEM95-null male mice. (**A**) Developmental outcomes following in vivo fertilization of WT eggs with WT, Hz or KO males. TMEM95-deficient males were unable to fertilize eggs, whereas no statistical differences were found in developmental rates following fertilization with sperm obtained from Hz or KO males (ANOVA p>0.05). (**B**) Developmental outcomes following in vitro fertilization of WT eggs with WT, Hz or KO males. TMEM95-deficient males were unable to fertilize eggs, whereas no statistical differences were found in developmental rates

*Figure 2 continued on next page*

*Figure 2 continued*

following fertilization with sperm obtained from Hz or KO males (ANOVA p>0.05).(**C**) Representative image of a non-fertilized egg following co-incubation with TMEM95-disrupted sperm. Egg penetration failure prevents zona hardening leading to the accumulation of sperm in the perivitelline space. (**D**) Motility parameters of WT, Hz or KO sperm analysed by CASA; no differences were found between groups in any of the parameters analysed (ANOVA p>0.05). (**E**) Transmission Electron Microscopy images of WT or KO sperm showing an overtly normal morphology in TMEM95-deficient sperm. (**F**) Average number of WT or KO sperm bound to each oocyte following binding assay. TMEM95 disruption did not impaired sperm binding to the egg membrane (ANOVA p>0.05). (**G**) Sperm-egg fusion assay. WT sperm fused with Hoechst pre-loaded zona-free eggs, which transferred the stain to them upon membrane fusion. In contrast TMEM95-depleted sperm were unable to fuse, exhibiting no Hoechst staining (only the egg DNA, marked by an arrow, is stained). (**H**) Development of WT eggs microinjected with WT or KO sperm following Intracytoplasmic Sperm Injection (ICSI). Similar developmental rates were obtained using WT or KO sperm (ANOVA p>0.05), indicating that TMEM95 null sperm were able to fertilize eggs when the sperm-egg membrane fusion step is bypassed by ICSI.

The online version of this article includes the following figure supplement(s) for figure 2:

**Figure supplement 1.** TMEM95 null sperm undergo acrosome reaction and bind to the egg membrane.

TMEM95 was involved in sperm binding by counting the number of KO or WT sperm bound to the egg membrane following sperm incubation with zona free eggs. The number of sperm bound to the egg membrane did not differ between both groups, suggesting that TMEM95 is dispensable for sperm binding to the oolema (*Figure 2F*). We also incubated zona-free eggs with two recombinant Cherry-tagged TMEM95 proteins, failing to detect TMEM95 binding to the egg membrane (*Figure 2—figure supplement 1B–C*). Together, these results indicate that TMEM95 is not involved on the initial binding but probably on post-binding events required for membrane fusion.

To further test the inability of TMEM95-disrupted sperm to fuse with the egg membrane, a sperm-egg fusion assay was performed preloading zona-free eggs with the DNA-binding dye Hoechst 33342 prior to fertilization with WT or KO sperm. In this assay, the stain loaded into the egg is transferred to the sperm only if gamete fusion occurs (*Inoue et al., 2005*). Despite the fact that multiple TMEM95-deficient sperm were bound to the oolema, none were stained by the dye stored in the eggs, confirming that sperm-egg membrane fusion does not occur in the absence of TMEM95 (*Figure 2G*). Subsequently, we tested whether gamete membrane fusion was the only process involved in the infertility of *Tmem95*-deficient male mice. TMEM95-null sperm were mechanically introduced inside the egg by intracytoplasmic sperm injection (ICSI), thereby artificially bypassing gamete membrane fusion. Eggs injected with sperm obtained from KO or WT males cleaved and developed at similar rates irrespective of the male genotype (*Figure 2G*). Heterozygous blastocysts obtained using KO sperm were transferred to pseudopregnant recipients, and the resulting pups were genotyped confirming that they were fathered by a *Tmem95*<sup>-/-</sup> male. Together, these results demonstrate that the fertility disruption caused by TMEM95 ablation can only be attributed to failure in sperm-egg fusion and thus, subsequent sperm incorporation into the ooplasm.

## TMEM95 does not interact with JUNO or IZUMO1

To assess whether TMEM95 was able to bind IZUMO1 or JUNO, we used an assay designed to detect extracellular protein interactions called avidity-based extracellular interaction screening (AVEXIS) (*Bushell et al., 2008*; *Kerr and Wright, 2012*). This assay detects direct interactions between receptor ectodomains expressed as soluble recombinant proteins in mammalian cells and uses highly-avid oligomeric forms to detect even very weak binding events that are a feature

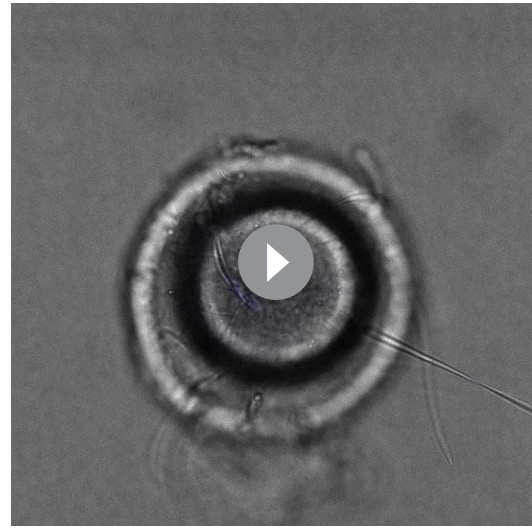

**Video 1.** Confocal z-sections of an egg fixed in PFA following IVF with TMEM95-null sperm. Multiple sperm are found in the perivitelline space, unable to fuse with the egg membrane.
https://elifesciences.org/articles/53913#video1

of this class of interactions (*Wright, 2009*). The entire ectodomains of JUNO, TMEM95 and IZUMO1 were expressed as monomeric biotinylated 'baits' and systematically tested for their ability to bind pentameric beta-lactamase tagged IZUMO1 and JUNO 'preys'. This approach confirmed the interaction of JUNO and IZUMO1 but no interaction between the extracellular regions of TMEM95 with either JUNO or IZUMO1 was detected (*Figure 3A*).

To further investigate the role of TMEM95 during fertilization, we developed a cellular fusion assay based on a split-GFP complementation approach. We first established two cell lines: one expressing mouse JUNO and the N-terminal fragment of GFP (GFP$_{1-7}$), and another cell line expressing mouse IZUMO1 and a C-terminal fragment of GFP (GFP$_{8-11}$) (*Figure 3—figure supplement 1A and B*). It is known that the two GFP fragments are able to reassemble spontaneously and complement fluorescence activity (*Kondo et al., 2010*; *Pedelacq and Cabantous, 2019*). The expression of functional JUNO and IZUMO1 on the surface of the cells was demonstrated by their ability to specifically bind to IZUMO1 and JUNO soluble protein probes respectively (*Figure 3B and C*; *Figure 3—figure supplement 1C, C', D and D'*). GFP$_{8-11}$ Izumo1 cells transiently transfected with an expression plasmid encoding the full length Tmem95 cDNA were mixed with the GFP$_{1-7}$ Juno cells and fusion events were scored by the presence of green cells (*Figure 3D*). By contrast to Syncytin-a transfected cells which served as a positive control, cells transfected with *Tmem95* did not induce any detectable cell fusion events even in the presence of JUNO and IZUMO1.

## Discussion

The search for candidate sperm proteins involved in fertilization has been particularly challenging and to date only another sperm protein, IZUMO1 was proved to be involved in this process (*Inoue et al., 2005*). Highlighting these difficulties, a recent article has reported that 30 genes that have a strong bias of expression in the testis are dispensable for reproduction (*Lu et al., 2019*). Also, on the sperm side, SPACA6 was found to play a role in sperm-egg membrane fusion based on the reproductive phenotype observed in BART97b mutant male mice. The random mutation harboured by these infertile males consists of a large insertion that introduced an 11 kb deletion disrupting *Spaca6* gene expression (*Lorenzetti et al., 2014*). During the revision of this manuscript, the essential role of SPACA6 on fertilization has been confirmed by targeted gene deletion by two independent groups (*Barbaux et al., 2020*; *Noda et al., 2020*). The discovery of TMEM95, a novel sperm protein essential for gamete fusion highlights that IZUMO1-JUNO interaction alone is not sufficient to mediate this process and suggests that more complex molecular interactions are required. Interestingly, IZUMO1-JUNO alone are not able to mediate fusion, as their interaction appears to mediate sperm-egg adhesion, not acting as fusogens (*Bianchi et al., 2014*).

The lack of interaction between TMEM95 and JUNO was an unexpected outcome, as the predicted structure of TMEM95 contains an α-helix equivalent to that found in IZUMO1 domain and the β-hairpin secondary structure, which provides the main platform for JUNO binding (*Ohto et al., 2016*). However, the residues on TMEM95 predicted β-hairpin differ from those present at IZUMO1 at the physicochemical level (*Figure 3—figure supplement 1*), suggesting that they can be responsible for JUNO recognition. Indeed, Trp-mediated interactions described for IZUMO1-JUNO binding ensure the conserved binding mode, but variable regions in the interface have been suggested to determine species-specificity (*Bianchi and Wright, 2015*; *Ohto et al., 2016*). It is possible that the structural domains shared by TMEM95 and IZUMO1 constitute a common feature of sperm proteins involved in fertilization, serving as mediators of protein-to-protein interactions. Another possibility would be that TMEM95 plays an architectural role on the sperm membrane similar to CD9. The mechanism by which the egg membrane protein CD9 intervenes in membrane fusion is not yet understood, with most evidence suggesting that it plays an architectural role that does not require a binding partner (*Chalbi et al., 2014*; *Jégou et al., 2011*). However, in contrast to CD9 ablation, which only causes complete infertility when combined with CD81 deficiency (*Rubinstein et al., 2006*), TMEM95 ablation alone completely blocks gamete fusion and thus, subsequent sperm entry into the ooplasm without generating any obvious structural defect in the sperm, thereby suggesting that TMEM95 may not play a major architectural role.

The role of TMEM95 in post-binding events may require one or several binding partners as, alone, it is unable to promote cell fusion following IZUMO1-JUNO binding in HEK293T cells. The residues located at TMEM95 predicted β-hairpin differing from those present in IZUMO1 β-hairpin

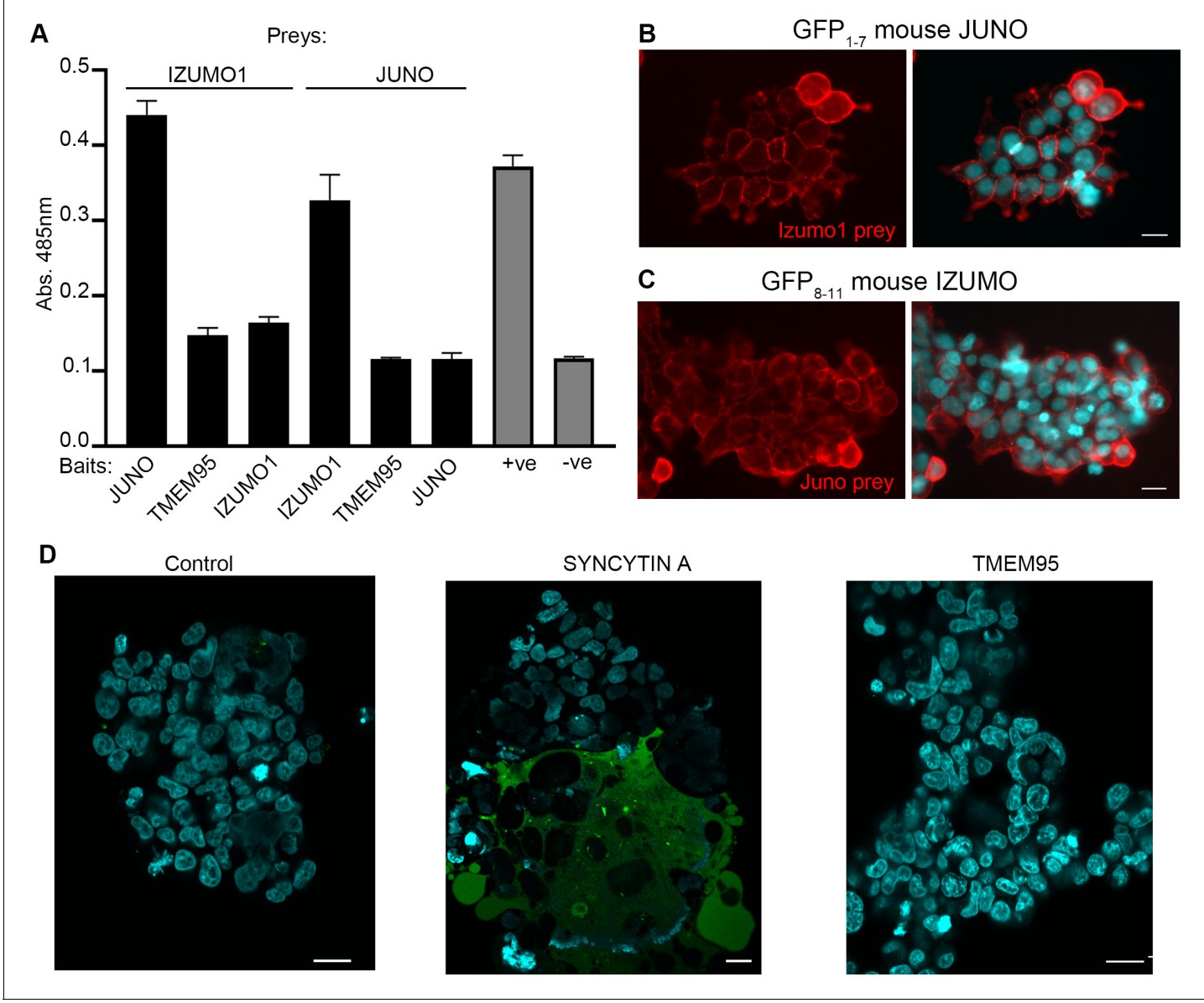

**Figure 3.** TMEM95 does not interact with JUNO nor IZUMO1. (**A**) Binding analysis using the AVEXIS assays shows that the soluble recombinant TMEM95 ectodomain does not interact with JUNO nor with IZUMO1. The entire ectodomains of the named proteins were expressed in HEK293-6E cells either as biotinylated baits or as pentameric beta-lactamase-tagged preys. Bait proteins were immobilised on streptavidin-coated plates and captured prey proteins quantified by measuring the absorbance of a colorimetric reaction product of the beta-lactamase substrate, nitrocefin. The CD200R (bait)-CD200 (prey) binding pair was used as positive control. The same prey, CD200R, was tested against TMEM95 and is shown as negative control. Bars represent means + s.d.; n = 3. (**B**) HEK293 cells stably expressing the N-terminal half of GFP (GFP1-7) and mouse JUNO stained with a highly avid IZUMO1 probe. (**C**) HEK293 cells stably expressing the C-terminal half of GFP (GFP8-11) and mouse IZUMO1 stained with a highly avid JUNO. (**D**) TMEM95 does not induce fusion when expressed in HEK293T cells in the presence of JUNO and IZUMO1 using a GFP-complementation cell fusion assay. HEK293T cells expressing either half of GFP and either JUNO or IZUMO1 were mixed and their fusogenic ability visualized by GFP fluorescence. The IZUMOI-expressing cells were either mock transfected prior to mixing (Control), transfected with *Syncitin a*, as a positive fusion control, or *Tmem95*. By contrast to the cells transfected with *Syncitin a*, *Tmem95* did not induce cell fusion. Cell nuclei are stained with DAPI and scale bar represents 20 μm.

The online version of this article includes the following figure supplement(s) for figure 3:

**Figure supplement 1.** Split GFP fusion assay.

**Figure supplement 2.** Sequence alignments of TMEM95 and IZUMO1.

could mediate affinity to other sperm or oocyte protein/s required for post-binding events. Given that the AVEXIS assay is performed exclusively using the extracellular domains of cell surface proteins, it is also possible that IZUMO1 and TMEM95 might interact via their transmembrane domains following IZUMO1-JUNO binding. While our manuscript was undergoing review, another article confirming our results on the essential role of TMEM95 in fertilization was submitted and published (*Noda et al., 2020*). This article uncovered the essential role of a sperm protein termed SOF1 on mammalian fertilization, suggesting that TMEM95, SPACA6 and SOF1 may function cooperatively with IZUMO1. However, the nature of this cooperation remains unclear, as the ablation of any of these three genes does not affect the amount and localization of IZUMO1 (*Barbaux et al., 2020*; *Noda et al., 2020* and our results). Besides, whereas co-immunoprecipitation analysis performed on cells overexpressing TMEM95, SPACA6, SOF1 and FIMP suggested interaction of these proteins with IZUMO1, this interaction could not be confirmed on testis lysates (*Noda et al., 2020*), and we failed to detect any IZUMO1:TMEM95 interaction on a high avidity binding assay capable of detecting subtle protein interactions. In contrast to IZUMO1, TMEM95, SPACA6 and SOF1 do not seem to exert a relevant role on sperm-oocyte binding (*Barbaux et al., 2020*; *Noda et al., 2020* and our results); therefore, their fundamental roles must be played in post-binding events. Together, these recent findings demostrate that gamete fusion is a complex fascinating puzzle involving proteins that have evolved for this specific task.

The discovery of the crucial function of TMEM95, a novel sperm membrane protein required for mammalian gamete fusion, provides a relevant insight into this process, arguably the most important membrane fusion event in a mammal´s life. This finding also paves the way for the investigation of the involvement of TMEM95 alterations in male infertility in humans and for the development of potential immune- or molecular-based contraceptive methods.

# Materials and methods

## Key resources table

| Reagent type (species) or resource | Designation | Source or reference | Identifiers | Additional information |
|---|---|---|---|---|
| Strain, strain background (*Mus musculus*) | Tmem95 KO | This article | | Generated by CRISPR (M and M), available upn request |
| Antibody | Rabbit Anti-TMEM95 | MyBioSource | MBS7004333 | |
| Antibody | Rabbit Anti-IZUMO1 | AbCam | Ab211623; RRID:AB_2650506 | |
| Antibody | Rat Anti-Izumo | Gift from Dr. Ikawa | Clone KS139-34 | |
| Antibody | Mouse Anti-Tubulin | Sigma | T8328, RRID:AB_1844090 | |
| Antibody | Rabbit anti-Flag | Sigma | F7425, RRID:AB_439687 | |
| Recombinant DNA reagent | Plasmid pMJ920 | (*Jinek et al., 2013*) | RRID:Addgene_42234 | |
| Recombinant DNA reagent | Plasmid px330 vector | (*Yang et al., 2014*) | RRID:Addgene_42230 | |
| Sequence-based reagent | Primers | This article | | Detailed on M and M (ordered from Sigma) |

## Animal models

All experimental procedures were approved by the INIA Animal Care Committee and Madrid Region Authorities (PROEX 040/17) in agreement with European legislation. *Tmem95* KO mice were generated by CRISPR technology as detailed below.

## In silico protein structure determination

Models were computed by the SWISS-MODEL server homology modelling pipeline (*Guex et al., 2009*; *Waterhouse et al., 2018*), which relies on ProMod3, an inhouse comparative modelling engine based on OpenStructure. The SWISS-MODEL template library (SMTL version 2019-03-20, PDB release 2019-03-15) was searched with BLAST and HHBlits for evolutionary related structures matching the target sequence TMEM95 (P0DJF3). The global and per-residue model quality was assessed using the QMEAN scoring function (*Benkert et al., 2011*); the score value of −3.11 was highlighted by a change of the 'thumbs-up' by the software.

sgRNA design and in vitro transcription sgRNA was designed against the first exon of *Tmem95* using bioinformatic tools to minimize the chances of offtarget genome edition (https://crispr.mit.edu). A sgRNA targeting the first exon of TMEM95 (ATTTCCACTTCGGCTTCGTC, NC_000077.6, Score 94) was synthesized and purified using Guide-it sgRNA In Vitro Transcription Kit (Takara). Capped polyadenylated Cas9 mRNA was generated by in vitro transcription (mMESSAGE mMA-CHINE T7 ULTRA kit, Life Technologies) using as template the plasmid pMJ920 (Addgene 42234) linearized with BbsI (NEB). mRNA was purified using MEGAClear kit (Life Technologies).

## Generation of *Tmem95* KO

C57CBAF1 female mice (7–8 weeks old) were superovulated by intraperitoneal injections of 5 IU of pregnant mare serum gonadotropin (PMSG, Folligon, MSD Animal Health) and an equivalent dose of human chorionic gonadotropin (hCG, Sigma) at a 48 hr interval. Superovulated female mice were mated with C57CBAF1 stud males and zygotes were recovered from oviducts.

Microinjections were performed with a micromanipulation system (Narishige MMO-202ND, MM-88) equipped on a Nikon Diaphot TMD inverted microscope. A mixture of 150 ng/μl of Cas9 mRNA and 50 ng/μl of sgRNA was delivered into the cytoplasm of zygotes (3–5 pl) using a filament needle (*Bermejo-Álvarez et al., 2015*).

Following microinjection, embryos were cultured in EmbryoMax KSOM Mouse Embryo Media (Millipore) at 37°C under 5% $CO_2$ for 4 days until they reached the blastocyst stage. Six blastocysts were transferred to a pseudopregnant Swiss recipient 2.5 days post-coitum (dpc), resulting in the birth of two pups: one male carrying in-frame mutations and one founder female carrying an in-frame indel and a frameshift allele (KO allele). This F0 female was crossed with C57BL/6 males to obtain heterozygous mutants. Heterozygous F1 individuals harbouring the KO allele were inter-crossed to produce WT, Hz or KO individuals used for the experiments.

## Mice genotyping

Genomic DNA samples from tail biopsies were prepared using FavorPrep Tissue Genomic DNA Extraction Mini Kit (Favorgen). PCR was performed using primers spanning the target sequence (F: 5′-CCCCCTTAGGATCCAGTGGT-3′, R: 5′-ACTCCTTCCATACCCCAGCA-3′, 255 bp product in WT) under the following conditions: 95°C for 2 min; 35 x (94°C for 20 s, 60°C for 30 s, 72°C for 30 s); 72°C for 5 min; hold at 8°C. The PCR products from F0 mice were purified using FavorPrep PCR Purification Kit (Favorgen), cloned into pMD20 T-vectors (Takara) using Blunt TA ligase (NEB), and transformed into DH5-α competent cells (ThermoFisher Scientific). Ten positive plasmid clones from each transformation were purified (Favorgen) and Sanger sequenced (Stabvida) to uncover the alleles generated following CRISPR-mediated edition harboured by each individual (*Lamas-Toranzo et al., 2019*).

Off-target analysis in the founder female was performed by analysing the 5 most probable off-target sequences detected by the sgRNA design bioinformatics tool (https://crispr.mit.edu). PCR were performed using the same conditions that above and the following primers: OT1 (F: 5′- ACTCTG TTCATCACCATGAGTCAA-3′, R: 5′- TGGCTTCCTTGTCTATGGTGT-3′, 264 bp product), OT2 (F: 5′-AAACCAATGAGATCCGCCGT-3′, R: 5′-TAGTTGCCAGGTTCGACCAC-3′, 223 bp product), OT3 (F: 5′- ACACACACGAGACTCGACAA-3′, R: 5′- TGCAAGATCTACACACGATCCA-3′, 233 bp product), OT4 (F: 5′- TGCAACAGAAGCAGGATGGT-3′, R: 5′- AACCCAGGCAGAAGAAGTGG-3′, 331 bp product) and OT5 (F: 5′- GTGCTGTGTCTGTTGCTTTT-3′, R: 5′- TGTAGTTTGGCCAGTTGTGC-3′, 220 bp product). PCR products were purified as above, Sanger sequenced and compared with the sequence of a WT individual of the same background.

Subsequent generations were genotyped by Sanger sequencing of PCR products until we developed a quantitative PCR high-resolution melting (qPCR-HRM) curve analysis that allowed us an easy detection of WT, Hz and KO individuals. qPCRs were performed using and Mic qPCR cycler (Bio Molecular Systems) with primers flanking the target sequence (F 5'-GGGAAGCCCATTCCTTCCAT-3', R 5'-GATCATGCCTTGGCAAACCG-3', 79 bp product in WT). Reaction conditions were as follows: 40x (94°C for 15 s, 56°C for 30 s, 72°C for 20 s). Melting curves were visualized using Mic qPCR hardware (Bio Molecular Systems) and contrasted with those obtained from known WT, Hz and KO samples confirmed by Sanger sequencing.

## Protein extraction and immunobloting

WT and KO sperm were collected from cauda epididymis in PBS supplemented with 0.1% PVP and centrifuged at 3,000 g for 7 min. Pellets and tissues samples (testis and accessory glands) were snap frozen in liquid nitrogen and kept at −80°C until analysis. For the analysis, frozen pellets were re-suspended in reducing SDS Sample Buffer (4X) (Millipore) and boiled for 10 min (lysis protocol #1) or in 1% Octyl β-D-Glucopyranoside (Sigma) solution in PBS and incubated on ice for 30 min (*Nishimura et al., 2011*). Cell suspensions from testis and accessory glands were obtained by thoroughly mincing tissues with razor blades. The cell pellet was re-suspended in 500 µl of 50 mM Tris-HCl ph 7.5, 1 mM EDTA, 1% Igepal, 0.1 mM PMSF, 10 mM iodoacetamide, 10 mM N-ethylmaleimide, phosphatase inhibitor and protease inhibitor, homogenized and incubated for 30 min on strong agitation at 4°C and then centrifuged at 20,000 g for 20 min at 4°C. After incubation, samples were centrifuged at 20,000 g for 10 min at 4°C, supernatants were separated in 16% SDS-PAGE and proteins were transferred to PVDF membranes. Subsequently, membranes were blocked with 5% BSA in TBST 1X for 1 hr at RT and incubated overnight in primary antibody solution in TBST 1 × 1% BSA. The primary antibodies used were anti-TMEM95 (1:1000 v/v, MyBioSource MBS7004333), anti-IZUMO1 (1:1000 v/v, Abcam ab211623) and anti-β-Tubulin (1:5000 v/v, Sigma T8328). Membranes were washed on the following day three times for 10 min with TBST 1X, incubated with corresponding peroxidase-conjugated secondary antibody and washed three times for 10 min with TBST 1X prior to visualization by chemiluminescence (Pierce ECL-Plus, Thermo Fisher Scientific).

The ~20 kDa band present on TMEM95 blot for WT samples but absent in KO samples was cut out and processed for proteomic analysis to confirm TMEM95 identity. Data processing was performed with Data Analysis program for LC/MSD Trap Version 3.3 (Bruker Daltonik) and Spectrum Mills MS Proteomics Workbench (Rev A.03.02.060B, Agilent Technologies) by the Molecular Biology Section, Service of Support to the Experimental Sciences (SACE), University of Murcia.

The bands in IZUMO and β-Tubulin blots were quantified by densitometric scanning and analysed with the ImageQuant TL software v2005 (GE Healthcare). Izumo1/Tubulin ratio was calculated in arbitrary units with data from sperm from 4 WT and 4 KO animals. The obtained data was subjected to Student's t-test and the level of significance was set at p<0.05. The software used was IBM SPSS Statistics (v22.0).

## Transcriptional analysis in reproductive tissues

Transcriptional analysis was performed as previously described (*Bermejo-Alvarez et al., 2010*). Briefly, total RNA was collected from testis, seminal vesicle, prostate and epididymis samples (3 samples/tissue) obtained from WT males using Trizol (Invitrogen). Following DNAse treatment (Promega), RNA was retrotranscribed (qScript, Quantabio) to cDNA (*Bermejo-Álvarez et al., 2015*). *Tmem95* and *Gapdh* transcripts were detected on cDNA by PCR using the amplification cycle described in the genotyping section. Primers to detect *Tmem95* were the same as those used for HRM-based genotyping (79 bp product). Primers used for *Gapdh* were F 5'-ACCCAGAAGACTG TGGATGG-3', R 5'-ATGCCTGCTTCACCACCTTC-3' (247 bp product). DNAse-treated non-retrotranscribed RNA obtained from testis served as negative control for DNAse treatment and PCR.

## Sperm immunocytochemistry

Sperm from WT or KO individuals was recovered from the cauda epididymis in PBS supplemented with 0.1% PVP. For IZUMO1 and TMEM95 relocalization analysis, sperm were acrosome reacted by 20 min incubation in HTF medium supplemented with 20 µM calcium ionophore. Following centrifugation (3,000 g for 7 min), sperm were fixed in 4% PFA in PBS for 5 min and washed twice in PBS.

Samples were then permeabilized with 0.1% Triton X-100 in PBS for 10 min, and blocked with 5% FCS in PBS for 45 min at 4°C. Next, samples were incubated with anti-TMEM95 antibody (1:100, MyBioSource) or anti-IZUMO1 antibody (1:100, a gift by Dr. Ikawa, clone KS139-34) overnight at 4°C. For double TMEM95 and IZUMO1 immunocytochemistry, samples were first incubated overnight with anti-TMEM95 followed by another overnight incubation with anti-IZUMO1. To determine TMEM95 localization within the acrosomal cap, the permeabilization step (10 min incubation in 0.1% Triton X-100) was skipped. As TMEM95, IZUMO1 and PNA were still detected in non-permeabilizing conditions, the fixation protocol was substituted with 30 min incubation in 4% PFA at 4°C (*Nishimura et al., 2011*), which still yielded similar results. Alexa Fluor 596 goat anti-rabbit IgG antibody (1:500, Invitrogen, for TMEM95) or Alexa Fluor 488 or 594 anti-rat IgG antibodies (1:500, Invitrogen, for IZUMO1) were used as a secondary antibody and incubation was performed for 2 hr at RT. Finally, samples were incubated for 5 min with 1 µg/ml FITC-PNA conjugate (Sigma) and 1 µg/ml Hoechst 33342 (Sigma). Samples were mounted and subsequently observed under an epifluorescence inverted microscope (Nikon Eclipse TE300) and NIS software (Nikon).

## In vivo fertilization analysis

C57CBAF1 female mice (7–8 weeks old) were superovulated as described above and mated with WT, Hz or KO individuals (4 individuals/group). Embryos were recovered from the oviduct on 1.5 dpc and cultured in vitro as described above. Cleavage rate was assessed on 1.5 dpc and blastocyst rate on 4.5 dpc. Statistical differences were analysed by ANOVA (SigmaStat package) and the level of significance was set at $p < 0.05$.

## In vitro fertilization (IVF)

Sperm from WT, Hz or KO individuals was recovered from the cauda epididymis in HTF medium and placed in the bottom of a previously equilibrated 300 µl drop of HTF covered with mineral oil for 2 hr prior to IVF. Following this pre-incubation time, the upper 150 µl of the drop were collected, and sperm concentration was analysed. Cumulus-oocytes complexes (COCs) were recovered from the oviducts of superovulated female mice 14 hr after hCG injection and placed in a 4-well dish with 400 µl of Human tubal fluid (HTF) medium in groups of ~40 COCs per well. Previously prepared sperm were immediately added to the well containing COCs at a final concentration of $10^6$ sperm/ml. After 4 hr of co-incubation, presumptive zygotes were sequentially washed in M2 and KSOM medium and cultured as described above. Statistical differences were analysed by ANOVA (SigmaStat package) and the level of significance was set at $p < 0.05$.

## Sperm motility analysis

Motility was analysed in sperm recovered from the cauda epididymis of WT, Hz and KO males (3 individuals/group). For CASA analysis, 20 µL of sperm suspension ($2 \times 10^6$ sperm/ml) were placed on a pre-warmed slide placed on a stage heated to 37°C and observed on an inverted microscope (Nikon Eclipse 50i) fitted with a digital camera (Basler A312f) capable of recording 25 frames/s. Five 1 s videos (20–60 moving sperm) were recorded in different fields and analyzed using the Integrated Semen Analysis System (ISAS). The parameters analysed (*Mortimer, 2000*) were straight-line velocity (VSL; time-averaged velocity of the sperm head along a straight line from its first position to its last position, expressed in µm/s); curvilinear velocity (VCL; time-averaged velocity of the sperm head along its actual curvilinear path, expressed in µm/s); average path velocity (VAP; velocity over an average path generated by a roaming average between frames, expressed in µm/s); linearity (LIN) (defined as (VSL/VCL)×100); straightness (STR) (defined as (VSL/VAP)×100); wobble (WOB) (defined as (VAP/VCL)×100); amplitude of lateral head (ALH) displacement (width of the lateral movement of the sperm head, expressed in µm) and beat cross frequency (BCF; number of times the sperm head crosses the direction of movement per second, expressed in Hz).

## Transmission electron microscopy analysis

WT and KO sperm were collected from the cauda epididymis in PBS supplemented with 0.1% PVP and centrifuged at 3000 g for 7 min. Pellet was resuspended in 2% glutaraldehyde in PBS and incubated for 2 hr at 4°C. After fixation, sperm were post-fixed in potassium ferrocyanide reduced osmium tetroxide for 1 hr. Following extensive washing, the samples were then dehydrated through

a graded series of ethanol and processed for embedding in Epon 812. Ultrathin sections were obtained with an ultramicrotome (Microm International GmbH) and mounted on formvar coated nickel grids. Ultrathin sections were counterstained with uranyl acetate followed by lead citrate and imaged in a Jeol JEL-1011 Transmission Electron Microscope.

## Sperm penetration and binding assays

Sperm penetration and sperm and TMEM95:Cherry egg binding assays were performed as described previously (*Bianchi et al., 2014*; *Inoue et al., 2005*). Briefly, COCs were collected from superovulated females 14 hr after hCG injection as described above. Cumulus cells were removed by incubation in 300 µg/ml hyaluronidase (Sigma) solution in M2 medium. Zona pellucida was removed by brief incubation in Acidic Tyrode´s medium. For the sperm penetration assay, zona-free mouse eggs were pre-incubated in HTF with Hoechst 33342 1 µg/ml for 10 min and washed before sperm addition. After 30 min of gametes co-incubation, the eggs were fixed in a 0.25% glutaraldehyde solution in PBS. For the sperm binding assay, eggs were also incubated for 30 min with sperm at a concentration of 1 million sperm/ml and stained with Hoechst after fixing; sperm from 3 KO and 3 WT males were tested on 12 eggs/male.

Two recombinant Cherry-tagged TMEM95 proteins were produced for protein-egg binding assay. Expression plasmids (pcDNA3.1[+]) were designed and constructed (GeneArt) to encode mouse TMEM95 protein (UniProt P0DJF3) fused to mCherry and FLAG tags as depicted in *Figure 2—figure supplement 1B*. TMEM95 was fused in-frame to mCherry, with mCherry inserted near the C-terminus of TMEM95 (TMEM95-Ccherry) or just downstream of the signal peptide of TMEM95 (TMEM95-Ncherry), separated in both cases by a 10 amino acid linker (GGGGSGGGGS). FLAG-tag was added to the C-terminus of both fusion proteins. Following DNA sequence verification, TMEM95-Ccherry and TMEM95-NCherry expression plasmids were amplified in DH5α competent cells (ThermoFisher Scientific) and purified using the GenEluted Plasmid Kit. Chinese Hamster Ovary cells (CHO cells, ECACC, The European Collection of Authenticated Cell Cultures) were grown (37°C, 5% $CO_2$ and 95% humidity) for 48–72 hr to 80–90% confluence using F-12 medium (Biowest) supplemented with 10% fetal bovine serum (Biowest) and 100 U/mL penicillin-streptomycin (Gibco). Transient transfections were performed with Lipotransfectina (Solmeglas). For each transfection, 4 µL Lipotransfectina transfection reagent were added to a final volume of 200 µL Opti-MEM reduced-serum medium (Gibco) pre-dissolved with 2 µg template plasmid and incubated for 15 min at room temperature (RT). The complex was diluted by adding 2 ml Opti-MEM and overlaid on growing cells (37°C, 5% $CO_2$ and 95% humidity). The medium containing the secreted proteins was collected after 48 hr, centrifuged at 4,000 g for 5 min at 4°C to remove cell debris, and concentrated in Vivaspin Turbo 4 of 10,000 Da (Sartorius). A final volume of 200–300 µL of concentrated proteins was obtained in 20 mM sodium phosphate buffer, pH 7.4 with protease inhibitor (EDTA-free EASYpack, Roche). Cell growing medium containing concentrated proteins was separated by SDS-PAGE and transferred to PVDF membranes which were probed with the primary antibody anti-Flag (Sigma F7425) at 1:1000 v/v in TBST 1X, 1% BSA, prior to visualization by chemiluminescence (Pierce ECL-Plus, Thermo Fisher Scientific). For the recombinant TMEM95:Cherry protein binding assay, zona-free eggs were exposed for 30 min to serial concentrations of recombinant proteins (0 to 0.9 mg/ml) diluted in protein-free medium; following incubation they were briefly washed in PBS and immediately observed under fluorescence microscopy. All specimens were observed under an epifluorescence inverted microscope (Nikon Eclipse TE300) and NIS software (Nikon).

## Intracytoplasmic sperm injection

Cauda epididymis sperm were collected in M2 medium from KO and WT males (3 individuals/group). Sperm was mixed with 5 volumes of a 10% solution of polyvinyl-pyrrolidone in M2. ICSI was performed in M2 medium at room temperature (*Fernández-Gonzalez et al., 2008*). Following sperm head injection, eggs were allowed to recover for 15 min in M2 at room temperature. Surviving eggs were cultured until the blastocyst stage in the same conditions as described above.

## Recombinant protein production and protein interaction screening by AVEXIS

The regions encoding the entire extracellular domains of IZUMO1 (Q9D9J7), Juno (Q9EQF4) and Tmem95 (P0DJF3) flanked by unique NotI and AscI sites were made by gene synthesis (Invitrogen, GeneArt Gene Synthesis) and cloned into protein expression vectors that encoded either biotinylated 'baits' or pentameric 'preys' tagged at the C-terminus with the beta-lactamase enzyme, the FLAG epitope and 6 histidines. All the ectodomains were expressed as soluble secreted proteins by transient transfections of HEK293-6E cells as described (*Bushell et al., 2008*). The rat Cd200 and Cd200R were used as positive controls.

Bait and prey proteins were normalized to activities suitable for the AVEXIS assay as previously described (*Kerr and Wright, 2012*). Biotinylated baits that had been dialyzed against HBS were immobilized on streptavidin-coated 96-well microtiter plates (Greiner Bio-One). Preys were then incubated for one hour, washed with HBS/0.1% Tween-20 three times and once with HBS. Hydrolysis of the beta-lactamase substrate Nitrocefin (Cayman Chemical) was quantified by reading absorbance values at 485 nm with a TECAN Spark plate reader. Nitrocefin was added at 125 µg/ml. The assay was repeated more than three times using independent protein preparations.

## Cell fusion assay: design of the expression vectors and establishment of stably transfected cell lines

To assess the ability of membrane proteins to induce cell fusion in vitro, we developed a complementation assay similar to the system described previously (*Kondo et al., 2010*) which exploits the ability of two GFP fragments to reassemble spontaneously and to reconstitute the fluorescent protein. We generated two cell lines each expressing one part of the GFP; one fragment is named $GFP_{1-7}$ and contains the first 7 beta sheets of the protein while the second fragment is named $GFP_{8-11}$ and is made of the last 4 beta sheets.

To force a quicker reassembly of the GFP fragments, in case it was needed, we exploited the oligomer-formation system inducible by rapamycin. We tagged FKBP1A and FRB to the split GFP because the two proteins interact with high affinity only in the presence of rapamycin (*Banaszynski et al., 2005*). The DNA encoding for FKBP1A linked to the N-terminus of $GFP_{1-7}$ and that of FRB linked to the C-terminus of GFP8-11 were obtained by gene synthesis (Invitrogen, GeneArt Gene Synthesis), digested and ligated into the pIRESHyg3 vector (Clontech) using NheI and BamHI restriction sites.

Clonal cell lines stably expressing the split GFP were established by transfecting HEK293T with the Lipofectamine 200 (ThermoFisher Scientific) and culturing them in selecting medium (61965–026 DMEM with Glutamax from ThermoFisher Scientific supplemented with 10% FBS and 250 µg/ml Hygromycin B). Single cells were sorted with the MoFlo XDP Cell sorter (Beckman Coulter), expanded and functionally tested. $GFP_{1-7}$ and $GFP_{8-11}$ cell clones were mixed together and transfected with a plasmid encoding for the mouse gene *syncytin A* (Origene untagged clone MC219753); the formation of green syncytia containing more than two nuclei was scored 48 hr after the transfection.

Mouse *Juno* was amplified by PCR from the cDNA of immature oocytes obtained from the cDNA library NIH_MGC_257_N (*Bianchi et al., 2014*). Mouse *Izumo1* was amplified from the ORF clone MG222708 (OriGene Technologies, Inc). Both were ligated in the pIRESPuro3 vector (Clonetech) using EcoRV-NotI and EcoRI-NotI restriction sites, respectively. Finally, the plasmid encoding *Juno* was transfected in the $GFP_{1-7}$ cell line and *Izumo1*-encoding plasmid was transfected in the $GFP_{8-11}$ cell line. Establishment of stable clones was obtained by selection with 10 µg/ml puromycin and single cell sorting. The expression of JUNO and IZUMO1 was assessed by staining with IZUMO1 and JUNO soluble recombinant pentamers, as previously described (*Bianchi et al., 2014*). Cells were incubated with the pentameric probes (preys) for 30 min at 37°C, fixed in 4% PFA and stained with an anti-FLAG monoclonal antibody Cy3 conjugated (A9594, clone M2, Sigma-Aldrich) for 1 hr at room temperature. Finally, cells were transferred to a microscope glass in Slowfade Gold mountant with DAPI (S36938 ThermoFisher Scientific) and images were acquired with the Leica TCS SP5 confocal microscope.

## Acknowledgements

IZUMO1 antibody was a generous gift by Dr Masahito Ikawa (Osaka University, Japan). Funding was provided by the Spanish Ministry of Science and Competitiveness (RYC-2012–10193, AGL2014-58739-R and AGL2017-84908-R to PBA, AGL2015-70159-P to MJM, RTI2018-093548-B-I00 to AGA and the network project AGL2016-71890-REDT), European Research Council (StG 757886-ELONGAN to PBA), Fundación Seneca-Agencia de Ciencia y Tecnología de Murcia (20887/PI/18 to MJM), the Irish Department of Agriculture, Food and The Marine (11/S/104), and the United Kingdom Medical Research Council (MR/M012468/1 to EB and GJW). ILT and SPC were supported by FPI and Ramón y Cajal contracts, respectively, from the Spanish Ministry of Science and Competitiveness. BFF is supported by a Marie Curie Fellowship from the European Commission.

## Additional information

### Funding

| Funder | Grant reference number | Author |
|---|---|---|
| Ministerio de Economía y Competitividad | RYC-2012-10193 | Pablo Bermejo-Álvarez |
| Ministerio de Economía y Competitividad | AGL2014-58739-R | Pablo Bermejo-Álvarez |
| Ministerio de Economía y Competitividad | AGL2017-84908-R | Pablo Bermejo-Álvarez |
| Ministerio de Economía y Competitividad | AGL2015-70159-P | María Jiménez-Movilla |
| Ministerio de Economía y Competitividad | RTI2018-093548-B-I00 | Alfonso Gutiérrez-Adán |
| Ministerio de Economía y Competitividad | AGL2016-71890-REDT | Alfonso Gutiérrez-Adán María Jiménez-Movilla Pablo Bermejo-Álvarez |
| H2020 European Research Council | StG 757886-ELONGAN | Pablo Bermejo-Álvarez |
| Fundación Séneca | 20887/PI/18 | María Jiménez-Movilla |
| Department of Agriculture, Food and the Marine | 11/S/104 | Pat Lonergan |
| Ministerio de Economía y Competitividad | FPI fellowship | Ismael Lamas-Toranzo |
| Ministerio de Economía y Competitividad | Ramón y Cajal contract | Serafín Pérez-Cerezales |
| European Union Seventh Framework Programme | Marie Curie fellowship | Beatriz Fernández-Fuertes |
| Medical Research Council | MR/M012468/1 | Enrica Bianchi Gavin J Wright |

The funders had no role in study design, data collection and interpretation, or the decision to submit the work for publication.

### Author contributions

Ismael Lamas-Toranzo, Writing - original draft, Writing - review and editing, Generation of KO model, breeding and genotyping of the colony, in vivo and in vitro fertility tests, immunocytochemistry, sperm fusion and binding assays; Julieta G Hamze, Writing - original draft, Writing - review and editing, Immunocytochemistry, Western-Blots, electron microscopy analysis, protein-egg binding assay; Enrica Bianchi, Formal analysis, Writing - original draft, Writing - review and editing, AVEXIS and cell fusion assays; Beatriz Fernández-Fuertes, Writing - review and editing, Immunocytochemistry; Serafín Pérez-Cerezales, Sperm motility analysis; Ricardo Laguna-Barraza, Raúl Fernández-González, Intra-cytoplasmic sperm injection; Pat Lonergan, Supervision, Funding acquisition, Writing -

review and editing; Alfonso Gutiérrez-Adán, Supervision, Funding acquisition; Gavin J Wright, Supervision, Funding acquisition, Writing - original draft, Writing - review and editing; María Jiménez-Movilla, Formal analysis, Supervision, Funding acquisition, Writing - original draft, Writing - review and editing, In silico protein structure determination, Western-Blots, electron-microscopy analysis, protein-egg binding assay, inter-species sequence analysis; Pablo Bermejo-Álvarez, Conceptualization, Resources, Formal analysis, Supervision, Funding acquisition, Writing - original draft, Writing - review and editing, Project coordinator, generation of KO model, off-target analysis, breeding and genotyping of the colony, in vivo and in vitro fertility tests, immunocytochemistry, transcriptional analysis, protein-egg binding assay

### Author ORCIDs
Ismael Lamas-Toranzo (iD) http://orcid.org/0000-0002-7790-2649
Gavin J Wright (iD) http://orcid.org/0000-0003-0537-0863
Pablo Bermejo-Álvarez (iD) https://orcid.org/0000-0001-9907-2626

### Ethics
Animal experimentation: All experimental procedures were approved by INIA Animal Care Committee and Madrid Region Authorities (PROEX 040/17) in agreement with European legislation.

### Decision letter and Author response
Decision letter https://doi.org/10.7554/eLife.53913.sa1
Author response https://doi.org/10.7554/eLife.53913.sa2

## Additional files
### Supplementary files
• Transparent reporting form

### Data availability
All data generated or analysed during this study are included in the manuscript and supporting files.

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
