## [Decision Letter]

**Acceptance summary:**

The manuscript reports an important discovery: the identification of the protein required for sperm and egg fusion, TMEM95. Spermatozoa deficient in TMEM95 are devoid of sperm-egg recognition and fail to fuse with the egg. Overall, given murine genetics and in vitro fertilization experiments presented in this manuscript, as well as previously reported data regarding the importance of TMEM95 for bovine fertility and human genetics, TMEM95 could emerge is an important sperm surface receptor required for fertilization. Therefore, the significance of this research is exceptionally high, not only for the basic science and reproductive biology but also for translational applications, such as fertility diagnostics and contraception research.

**Decision letter after peer review:**

Thank you for submitting your article "TMEM95 is required for sperm-oocyte membrane fusion" for consideration by *eLife*. Your article has been reviewed by three peer reviewers, including Polina V Lishko as the Reviewing Editor and Reviewer #1, and the evaluation has been overseen by Anna Akhmanova as the Senior Editor.

The reviewers have discussed the reviews with one another and the Reviewing Editor has drafted this decision to help you prepare a revised submission.

While the goal is to provide the essential revision requirements as a single set of instructions, in this particular case we will provide a complete set of the reviews, so you can address all concerns as it is essential for the revision that is necessary for us to publish your work.

Summary:

The manuscript by Lamas-Toranzo et al., reports an important discovery: the identification of the protein required for sperm and egg fusion, TMEM95. Spermatozoa deficient in TMEM95 lacks sperm-egg recognition/fusion abilities much similar to IZUMO1-disrupted spermatozoa, and its requirement in the process is documented in this manuscript. However, the paper lacks detailed evidence regarding exactly how TMEM95 contributes to the process. Additionally, one of the major claims in the manuscript (absence of interaction between TMEM95 and other sperm/egg binding proteins) is not supported by the data. These data are omitted from the manuscript. Overall, given murine genetics and IVF experiments presented in this manuscript, as well as previously reported data regarding the importance of TMEM95 for bovine fertility and human genetics, TMEM95 could emerge is an important sperm surface receptor required for fertilization. Therefore, the significance of this research is exceptionally high, not only for the basic science and reproductive biology but also for translational applications, such as fertility diagnostics and contraception research.

Essential revisions:

The reviewers raise a number of concerns that must be adequately addressed before the paper can be accepted. Some of the required revisions will likely require further experimentation within the framework of the presented studies and techniques.

Specifically:

1) Experiments showing that TMEM95 recombinant protein acquires appropriate physiological activity and retains the ability to bind to the zona-free egg surface directly must be performed.

2) Immunoprecipitation experiments with sperm lysates using anti-IZUMO1 and/or anti-TMEM95 antibodies need to be shown.

3) While it was shown that TMEM95-disrupted spermatozoa express an IZUMO1 protein, it is unclear whether IZUMO1-disrupted sperm still possess a TMEM95 protein.

4) Additional characterization of TMEM95 expression in testes, epididymis, accessory glands etc on the protein level), as well as the tissue specificity should be performed.

5) The authors should carry out the localization studies in permeabilized and non-permeabilized intact WT sperm in order to detect in which exact cellular compartment TMEM95 is found.

6) The analysis of the number of sperm that bind to the egg surface should be performed.

7) As suggested all experiments must be performed with positive controls that confirm that the technique employed. Data not shown is not permitted.

8) Many parts of the manuscript (Abstract, Title, Results and Discussion, as well as terminology used: "egg" vs "oocyte") need to be rewritten and clarified as suggested by reviewers. Overall, the manuscript would benefit from thorough language editing, as well as additional data and experiments.

Reviewer #1:

The manuscript by Lamas-Toranzo et al., reports an important discovery: the identification of the protein required for sperm and egg fusion, TMEM95. The data provided are of good quality, and in vitro fertilization, experiments are very convincing. However, one of the major claims in the manuscript (absence of interaction between TMEM95 and other sperm/egg binding proteins) is not supported by the data. These data were either accidentally omitted from the manuscript or were not intended to be shown. Overall, given murine genetics and IVF experiments presented in this manuscript, as well as previously reported data regarding the importance of TMEM95 for bovine fertility and human genetic data, this reviewer has confidence that TMEM95 is an important sperm surface receptor required for fertilization. Therefore, the significance of this research is exceptionally high, not only for the basic science and reproductive biology but also for translational applications, such as fertility diagnostics and contraception research. However, the manuscript is less polished in regards to its writing style, the data representation, and overall would benefit from thorough language editing.

1) The first paragraph of the Introduction and throughout the text. Sperm usually do not fertilize an oocyte, they fertilize an egg or ovum. An oocyte is an immature female gamete still inside the follicle (for mammals), once it ovulates, it becomes the egg/ovum. Hence, terminology, such as "sperm-oocyte fusion" is technically incorrect and confusing, and should be replaced with sperm-egg fusion, including in the title.

2) Introduction: the last paragraph describing bovine genetics seems to be out of place. The authors first describe gene-editing in mice, and immediately jump to bovine genetics- it is hard to follow why such a transition was made.

3) Figure 1D: the entire western blot for TMEM95 should be shown to evaluate the specificity of the antibody used in the study.

4) Figure 3. The authors claim that the pull-down assay was performed to explore whether TMEM95 interacts with either IZUMO or Juno. However, the only experimental data that are actually shown is either heterologous expression of TMEM and its truncated form in CHO cells or the pull-down assays probed with anti-TMEM antibody. There is NO panel that shows western blot with either anti-IZUMO or anti-Juno antibodies. Moreover, membrane protein extraction from mature sperm is a difficult task, since most of these proteins are covalently attached to the underlying cytoskeletal structures. To rule out TMEM95/IZUMO/Juno interaction, authors must express all three heterologous in CHO cells, and then perform pull-down assay to explore their potential interaction.

5) Introduction first paragraph "perivitellin", should be perivitelline.

6) A triplicate of pull-down experiments is needed.

Reviewer #2:

Through the gene-modified animals, IZUMO1 and SPACA6 on the sperm side, and JUNO (IZUMO1R) and CD9 on the ovum side, have been identified as essential factors for sperm-egg fusion. In this study, the authors newly found a third sperm factor, TMEM95, which was originally identified by genome-wide screening from subfertile bulls, and is indispensable for gamete fusion in mice as well. Spermatozoa lacking TMEM95 are all missing of sperm-egg recognition/fusion abilities much similar to IZUMO1-disrupted spermatozoa, and its requirement in the process is carefully documented in this manuscript.

However, I think that the paper lacks detailed evidence regarding exactly how TMEM95 contributes to the process. Thus, in my opinion, this manuscript in its current form is not yet ready for publication in *eLife*.

The authors have concluded that TMEM95 is not involved in the IZUMO1-JUNO system due to results obtained via combination of co-immunoprecipitation (His-tag pull down) and HPLC-MS/MS analyses, however, I am concerned that these experiments have not been validated sufficiently. Particularly, I doubt if TMEM95 recombinant protein acquires appropriate physiological activity. Does this recombinant have the ability to bind to the zona-free oocyte surface directly? Also, in terms of IZUMO1, did you perform immunoprecipitation experiments with sperm lysate using anti-IZUMO1 and/or anti-TMEM95 antibodies? In addition, Figure 1D shows that TMEM95-disrupted spermatozoa possess an IZUMO1 protein; however, conversely, does IZUMO1-disrupted spermatozoa possess a TMEM95 protein? Mutual experiments are required to elucidate the detailed molecular relationship between these factors.

I think that those verifications need to be addressed.

Reviewer #3:

The manuscript by Lama-Toranzo et al. describes the identification of TMEM95, a sperm protein essential for mouse gamete fusion. Considering that, so far, only one sperm protein (i.e. IZUMO) has been found to be critical for this process, these results indicate that TMEM95 and IZUMO are both necessary but not sufficient for gamete fusion, representing a very interesting contribution to the field of mammalian fertilization. Nevertheless, the high structural similarity between TMEM95 and IZUMO 1 domain together with the same localization of these two proteins in intact and acrosome-reacted sperm supports the idea that TMEM95 likely participates in gamete fusion through the same ligand-receptor mechanisms involved in IZUMO-JUNO mediated gamete fusion. In this regard, the experiments showing that TMEM95 does not interact with JUNO are not convincing as they lack of appropriate positive controls. Moreover, considering that the only successful approach to identify JUNO has involved the use of a pentameric IZUMO protein due to the very low affinity between the two molecules (Bianchi and Wright et al., 2016), it is not clear why the authors analyzed TMEM95-JUNO interaction using just a regular co-immunoprecipitation assay. Thus, from a mechanistic point of view, the results on the participation of TMEM95 in gamete fusion does not seem to provide novel information. The molecular mechanisms underlying TMEM95 role in gamete fusion should be better analyzed and discussed.

There are also several other points that the authors should take into consideration before the manuscript can be accepted for publication in *eLife*.

Specific comments:

Abstract:

1) The authors mentioned that "TMEM95-deficient sperm were unable to fuse with or penetrate the oocyte membrane". Sperm bind to and then fuse with the egg plasma membrane and, finally, they penetrate into (or are incorporated into) the ooplasm. It is not correct to say that sperm cannot "penetrate" the oocyte membrane. This expression should be modified here as well as in other sections of the manuscript.

2) It is not clear which is the meaning of "kinetically" normal sperm. Are the authors referring to sperm motility and/or hyperactivity? This should be more clearly indicated.

3) A final sentence briefly indicating the significance of the findings should be added at the end of this section.

Results:

4) the authors should include in this section additional information regarding the characterization of TMEM95 protein which could contribute to a better interpretation of the results obtained. i.e. tissue expression (testes, epididymis, accessory glands etc) and tissue specificity.

5) Based on immunocytochemistry studies, the authors indicate that TMEM95 is present in the acrosome membrane of WT intact sperm. They should be aware that acrosome membrane is not the same than acrosomal region. If sperm are intact, the presence of TMEM95 in the acrosome membrane would imply this is an internal rather that a superficial protein. The authors should carry out the localization studies in permeabilized and non-permeabilized intact WT sperm in order to discriminate between the two possibilities.

Figure 1E legend indicates that "TMEM95 is localized around the acrosomal region of acrosome-intact sperm and in the sperm head after the acrosome reaction". The term "sperm head" is very vague. The authors should indicate where in the sperm head TMEM95 is localized. According to Figure 1F, TMEM95 should be located in the equatorial segment but the image in Figure 1E does not seem to show that localization. In addition to this, Figure 1E does not include any description about what the arrows are showing in each case.

The authors indicate that after the acrosome reaction, TMEM95 translocates to the equatorial segment. A protein can relocalize to the equatorial segment as a consequence of either its exposure due to the release of decapacitation factors or its migration from the acrosomal region. Based on the studies included in this manuscript, the authors cannot state that TMEM95 "translocates" from one region to the other.

6) In addition to the evaluation of gamete fusion by the Hoechst technique, the authors should analyze the number of sperm that bind to the egg surface. This analysis will provide information on whether or not TMEM95 participates in the first stage of gamete fusion i.e. sperm-egg binding

7) Figure 3: as mentioned above, the co-immunoprecipitation assay lacks appropriate positive controls that confirm that the technique employed is in fact capable of detecting a potential interaction between TMEM95 and JUNO (i.e a control showing IZUMO/JUNO interaction).

Figure 3 / Table 2: the authors carried out a series of Pull-Down experiments to identify potential TMEM95 partners in either the egg or the sperm. However, the explanation of these experiments and the interpretation of the results obtained are very poorly described.

Discussion:

The discussion about the potential molecular mechanisms underlying TMEM95 role in gamete fusion is rather confusing. The authors first suggest that TMEM95 may not require an oocyte partner. Then, they propose TMEM95 may have affinity to a yet undescribed receptor and then, based on the fact that TMEM95 acts independently from JUNO and IZUMO1 (which is not really demnstarted), they propose that TMEM95 may play an essential role rather than by single protein-to-protein interactions. The authors should more clearly conclude whether their observations support that TMEM95 participates in the first stage of gamete fusion (i.e sperm-egg binding) through a ligand-receptor mechanism or whether the protein could be acting as a fusogenic molecule capable of being intercalated into the egg lipid bilayer.

[Editors' note: further revisions were suggested prior to acceptance, as described below.]

Thank you for resubmitting your work entitled "ITMEM95 is a sperm protein essential for mammalian fertilization" for further consideration by *eLife*. Your revised article has been evaluated by Anna Akhmanova (Senior Editor) and a Reviewing Editor.

The manuscript has been significantly improved but there are some remaining issues that need to be addressed before acceptance, as outlined below. We understand that due to the current pandemic situation additional experiments might not be feasible, therefore, we ask authors to focus on the textual revisions as suggested by reviewers. Here we provide a full review as it contains many suggestions authors might find helpful.

Reviewer #1:

The authors addressed my concerns and the revision resulted in an improved manuscript.

Reviewer #2:

I am almost satisfied with the modifications provided by the authors, however, I still remain some concerns before publication in *eLife*.

1) The authors should mention SPACA6 as a gamete fusion related factor not only in Discussion but also in Abstract and Introduction, because I think SPACA6 is an unneglectable factor for this process (Barbaux et al., 2020).

2) In Figure 2—figure supplement 1C, there is no positive control (e.g. IZUMO1) in this system.

3) I recommend a rat anti-mouse IZUMO1 monoclonal antibody clone 125 (abcam, ab211626 or BioAcademia, 73-045 or anticorps-eligne, ABIN2452040) for detection in western blotting instead of polyclonal antibody because of high reliability.

Reviewer #3:

The revised version submitted by Lozano et al. includes new important studies that address the two main concerns raised by this (and other) reviewers i.e. 1) whether or not TMEM95 interacts with IZUMO1 and/or Juno, and 2) the mechanism underlying TMEM95 involvement in gamete fusion.

Regarding the first point, the authors decided to use the same strategy that had been employed to identify Juno, the partner of IZUMO, having now the appropriate controls. Moreover, the studies were performed in collaboration with the researchers that reported the identification of Juno who are now co-authors of the paper. In contrast to the previous immunoprecipitation and pull-down experiments, this new study clearly shows that TMEM95 interacts neither with IZUMO1 nor with Juno, representing a key incorporation to the paper.

To investigate the mechanisms underlying TMEM95 involvement in gamete fusion i.e whether the protein participates in the first stage of sperm binding to the oolemma or in the subsequent stage of membrane fusion itself, the authors 1) carried out an in vitro assay to evaluate the ability of TMEM95 mutant sperm to interact with the egg plasma membrane observing no differences with the controls, 2) analyzed the ability of TMEM95 to interact with the surface of zona-free eggs, detecting no binding of the protein to the egg, and 3) performed transfection studies to analyze the ability of TMEM95 to promote membrane fusion, observing no detectable fusion events in the analyzed cells. According to these observations, the authors concluded that TMEM95 does not participate in sperm-egg binding neither is a fusogenic protein and proposed that the protein must be involved in a post sperm-egg binding event previous to membrane fusion, as reported for IZUMO.

These studies have certainly fully answered the two main concerns of this reviewer supporting the publication of the manuscript in the journal. Nevertheless, there are some aspects that the authors should take into consideration in order to improve the quality of the manuscript.

– The authors indicated that the exact localization of the protein (internal or external) could not be analyzed because the employed fixation itself produced a permeabilization of the cells. Whereas this is true for many fixation conditions, the authors should consider carrying out immunolocalization studies in live, non-fixed sperm in suspension. This is an easy approach that will allow the authors to define the protein localization in the cell. This is not a minor point as it is known that the localization of a molecule provides important information regarding the mechanisms underlying its functional role, an issue still not fully resolved for TMEM95. In this regard, the author's claim that TMEM95 likely localizes to the acrosomal membrane because both it relocalizes to the equatorial segment after the acrosome reaction and it has a TM domain are not correct as plasma membrane proteins with or without TM domains do so as well.

– Whereas the authors have carried out the tissue expression and specificity studies requested by this reviewer, I could not find the description of the results obtained within the text but just in the figure. This should be corrected.

– I would remove the first section within Results entitled "TMEM95 protein in silico folding suggests a role in gamete fusion" as it describes just an in silico analysis that does not really provide any new information compared to the bull protein and does not deceives an independent paragraph. I would join the results of this paragraph that describes just Figure 1A with the results of the following paragraph describing the rest of Figure 1. The authors may need to change the title of that second paragraph too.

– The authors indicated that the lack of TMEM95 interaction with IZUMO together with the normal relocalization of IZUMO1 in TMEM95 KO sperm makes it unlikely that IZUMO1 KO would lead to the ablation of TMEM95 and that performing that experiment would delay the publication of these findings. In this regard, I think the authors could contemplate the possibility of sending their anti-TMEM95 antibody to Dr Ikawa who could perform this easy localization study in his lab. Of note in this regard, the authors already have contact with Dr Ikawa's lab which has provided the anti-IZUMO antibody to them.

– In my opinion, there are several observations which are not sufficiently discussed. The authors may speculate about several issues such as the finding that TMEM95 and IZUMO are both necessary but not sufficient for gamete fusion, how the protein reaches the equatorial segment, potential ways to identify TMEM95 partners, whether TMEM95 is present or not in IZUMO KO cells, among others, which will certainly enrich the discussion.

---

## [Author Response]

Essential revisions:The reviewers raise a number of concerns that must be adequately addressed before the paper can be accepted. Some of the required revisions will likely require further experimentation within the framework of the presented studies and techniques.Specifically:1) Experiments showing that TMEM95 recombinant protein acquires appropriate physiological activity and retains the ability to bind to the zona-free egg surface directly must be performed.

We have conducted new experiments to determine a possible role of TMEM95 in sperm binding to the egg, concluding that TMEM95 is dispensable for sperm binding based on the following evidence: 1) TMEM95 ablation did not reduce sperm binding to the oocyte (sperm binding assay, Essential revision 6, Figure 2F), and 2) TMEM95:Cherry proteins did not bind to zona-free eggs (Figure 2—figure supplement 1C). These results suggest that TMEM95 is involved in post-binding events required for fertilization.

2) Immunoprecipitation experiments with sperm lysates using anti-IZUMO1 and/or anti-TMEM95 antibodies need to be shown.

We have substituted the previous pull-down assay with an assay designed to detect extracellular protein interactions (AVEXIS), which was the one responsible for the discovery of IZUMO1-JUNO interaction. Neither pull-down nor immunoprecipitation assays are able to detect such a subtle interaction (Figure 3A). This highly sensitive assay failed to detect any interaction between TMEM95 and JUNO or IZUMO.

3) While it was shown that TMEM95-disrupted spermatozoa express an IZUMO1 protein, it is unclear whether IZUMO1-disrupted sperm still possess a TMEM95 protein.

We believe that given TMEM95 is not closely related to IZUMO1 given that: 1) in the absence of TMEM95, IZUMO1 relocates normally from the acrosomal cap to the equatorial section following the acrosome reaction (Figure 1—figure supplement 1B), 2) IZUMO1 expression is not affected by TMEM95 disruption (Figure 1D), 3) TMEM95 does not interact with IZUMO1 (Figure 3A), and 4) TMEM95 does not interact with JUNO (Figure 3A). Therefore, there is no solid rationale to believe that IZUMO1 disruption would lead to TMEM95 disruption. Generating a colony of IZUMO1 KO to test an improbable co-disruption of TMEM95 would take ~6 months, an excessive and, we believe unnecessary delay to reporting our findings.

4) Additional characterization of TMEM95 expression in testes, epididymis, accessory glands etc on the protein level), as well as the tissue specificity should be performed.

We have performed new experiments analysing TMEM95 expression at mRNA and protein levels observing that the gene is expressed exclusively to the testis (Figure 1—figure supplement B-C).

5) The authors should carry out the localization studies in permeabilized and non-permeabilized intact WT sperm in order to detect in which exact cellular compartment TMEM95 is found.

We have conducted new experiments to determine TMEM95 localization within the acrosomal region of intact WT sperm. Previous experiments showed that TMEM95 was present in the equatorial segment of acrosome reacted sperm, and in the acrosomal cap of acrosome intact sperm, as the TMEM95 signal co-localized with PNA (Figure 1E-F). We have now performed IHC in WT sperm following permeabilizing and non-permeabilizing conditions as requested, observing TMEM95, IZUMO1 (acrosome membrane marker, Inoue et al., 2012) and PNA (internal acrosomal marker, Lybaert et al., 2009 Histol. Histopathol.) signals in both conditions (Figure 1—figure supplement 3A). This result indicates that fixation did not prevent antibody access into the acrosome in the absence of permeabilization (i.e., exposure to Triton X-100) during the whole IHC procedure. Further attempts to “seal” sperm and acrosome membranes, including increasing fixation time in 4 % PFA to 30 min at 4 ºC (following the protocol of Nishimura et al., 2011), were also unsuccessful: both IZUMO1 and PNA signals were clearly visible. These results suggest that signal detection following IHC in non-permeabilizing conditions may not be a solid criteria to exclude acrosomal localization. In agreement, whereas absence of IHC signal following non-permeabilizing conditions proved the intracellular localization of a protein in sperm, the opposite (i.e., IHC signal in non-permeabilizing conditions proving external membrane localization) is not conclusive: PNA, which specifically bounds to the acrosomal content, was also detected following non-permeabilizing conditions (Nishimura et al., 2011). We do not have solid evidence to claim that TMEM95 localizes to the acrosomal membrane, albeit that may be a likely location given 1) its co-localization with PNA and IZUMO1 on intact sperm, 2) its relocalization following acrosome reaction and 3) the presence of a transmembrane domain. We have modified the text to state that it localizes to the acrosomal cap in acrosome intact sperm.

6) The analysis of the number of sperm that bind to the egg surface should be performed.

We have performed this analysis observing no difference in sperm binding to the egg surface in the absence of TMEM95 (Figure 2F), suggesting that TMEM95 is not required for sperm binding.

7) As suggested all experiments must be performed with positive controls that confirm that the technique employed. Data not shown is not permitted.

We have included positive controls.

8) Many parts of the manuscript (Abstract, Title, Results and Discussion, as well as terminology used: "egg" vs "oocyte") need to be rewritten and clarified as suggested by reviewers. Overall, the manuscript would benefit from thorough language editing, as well as additional data and experiments.

We have extensively rewritten the manuscript and hope that is now suitable for publication.

Reviewer #1:[…] 1) The first paragraph of the Introduction and throughout the text. Sperm usually do not fertilize an oocyte, they fertilize an egg or ovum. An oocyte is an immature female gamete still inside the follicle (for mammals), once it ovulates, it becomes the egg/ovum. Hence, terminology, such as "sperm-oocyte fusion" is technically incorrect and confusing, and should be replaced with sperm-egg fusion, including in the title.

We used oocyte and ovulated cumulus-oocyte-complex (COC) as they are commonly employed terms in mammalian embryology literature. In order to avoid any confusion for a wider audience, we have changed “oocyte” to “egg” throughout the manuscript.

2) Introduction: the last paragraph describing bovine genetics seems to be out of place. The authors first describe gene-editing in mice, and immediately jump to bovine genetics- it is hard to follow why such a transition was made.

We wanted to acknowledge these references early in the manuscript. We have moved them to the beginning of the Results section in the revised version.

3) Figure 1D: the entire western blot for TMEM95 should be shown to evaluate the specificity of the antibody used in the study.

WB images are now shown (Figure 1—figure supplement 2). TMEM95 WB show unspecific bands but a band of the expected size is present in WT samples and absent in KO samples. Peptide sequencing of that band confirmed TMEM95 identity (Table 1).

4) Figure 3. The authors claim that the pull-down assay was performed to explore whether TMEM95 interacts with either IZUMO or Juno. However, the only experimental data that are actually shown is either heterologous expression of TMEM and its truncated form in CHO cells or the pull-down assays probed with anti-TMEM antibody. There is NO panel that shows western blot with either anti-IZUMO or anti-Juno antibodies. Moreover, membrane protein extraction from mature sperm is a difficult task, since most of these proteins are covalently attached to the underlying cytoskeletal structures. To rule out TMEM95/IZUMO/Juno interaction, authors must express all three heterologous in CHO cells, and then perform pull-down assay to explore their potential interaction.

We have now conducted new experiments which demonstrate that TMEM95 does not bind to either IZUMO1 or JUNO using the same technique (AVEXIS, Kerr and Wright, 2012) that was used to identify IZUMO1 and JUNO interaction (Bianchi et al., 2014, Figure 3A). Given the absence of interaction we hypothesized that TMEM95 may be required to induce fusion following IZUMO1 and JUNO interaction. To test this possibility we have performed a GFP-complementation cell fusion assay in HEK293T cells observing that TMEM95 expression did not induce fusion on cells expressing JUNO and IZUMO (Figure 3D). These results suggest that other proteins may be required to achieve cell fusion.

5) Introduction first paragraph "perivitellin", should be perivitelline.

Changed, thank you for the correction.

6) A triplicate of pull-down experiments is needed.

As mentioned above, we have substituted those experiments by AVEXIS, a more sensitive protein interaction assay, performed in triplicate.

Reviewer #2:[…] The authors have concluded that TMEM95 is not involved in the IZUMO1-JUNO system due to results obtained via combination of co-immunoprecipitation (His-tag pull down) and HPLC-MS/MS analyses, however, I am concerned that these experiments have not been validated sufficiently. Particularly, I doubt if TMEM95 recombinant protein acquires appropriate physiological activity. Does this recombinant have the ability to bind to the zona-free oocyte surface directly? Also, in terms of IZUMO1, did you perform immunoprecipitation experiments with sperm lysate using anti-IZUMO1 and/or anti-TMEM95 antibodies? In addition, Figure 1D shows that TMEM95-disrupted spermatozoa possess an IZUMO1 protein; however, conversely, does IZUMO1-disrupted spermatozoa possess a TMEM95 protein? Mutual experiments are required to elucidate the detailed molecular relationship between these factors. I think that those verifications need to be addressed.

As detailed in the general comments above, we have conducted new experiments (protein interaction and sperm-egg binding assays) concluding that TMEM95 is not directly involved in binding. Using a highly sensitive protein interaction assay able to detect IZUMO1-JUNO interaction (i.e., more sensitive than immunoprecipitation or pull-down assays), we have observed that IZUMO1 does not interact with TMEM95. This new evidence together with the normal relocalization of IZUMO1 in TMEM95 KO sperm makes it unlikely that IZUMO1 KO would lead to the ablation of TMEM95. Performing that experiment would lead to an unjustified delay in the publication of these findings.

Reviewer #3:The manuscript by Lama-Toranzo et al. describes the identification of TMEM95, a sperm protein essential for mouse gamete fusion. Considering that, so far, only one sperm protein (i.e. IZUMO) has been found to be critical for this process, these results indicate that TMEM95 and IZUMO are both necessary but not sufficient for gamete fusion, representing a very interesting contribution to the field of mammalian fertilization. Nevertheless, the high structural similarity between TMEM95 and IZUMO 1 domain together with the same localization of these two proteins in intact and acrosome-reacted sperm supports the idea that TMEM95 likely participates in gamete fusion through the same ligand-receptor mechanisms involved in IZUMO-JUNO mediated gamete fusion. In this regard, the experiments showing that TMEM95 does not interact with JUNO are not convincing as they lack of appropriate positive controls. Moreover, considering that the only successful approach to identify JUNO has involved the use of a pentameric IZUMO protein due to the very low affinity between the two molecules (Bianchi and Wright et al., 2016), it is not clear why the authors analyzed TMEM95-JUNO interaction using just a regular co-immunoprecipitation assay. Thus, from a mechanistic point of view, the results on the participation of TMEM95 in gamete fusion does not seem to provide novel information. The molecular mechanisms underlying TMEM95 role in gamete fusion should be better analyzed and discussed.

We agree that the conventional pull-down assay performed may fail to detect a subtle TMEM95-Juno interaction. This pull-down assay performed with recombinant proteins is more sensitive than immunoprecipitation, but still may still fail to uncover subtle protein interactions. We have now used the same highly sensitive protein interaction assay employed by Bianchi et al., 2014, proving that TMEM95 does not bind to JUNO or IZUMO1.

There are also several other points that the authors should take into consideration before the manuscript can be accepted for publication in eLife.Specific comments:Abstract:1) The authors mentioned that "TMEM95-deficient sperm were unable to fuse with or penetrate the oocyte membrane". Sperm bind to and then fuse with the egg plasma membrane and, finally, they penetrate into (or are incorporated into) the ooplasm. It is not correct to say that sperm cannot "penetrate" the oocyte membrane. This expression should be modified here as well as in other sections of the manuscript.

We have corrected the Abstract as suggested.

2) It is not clear which is the meaning of "kinetically" normal sperm. Are the authors referring to sperm motility and/or hyperactivity? This should be more clearly indicated.

We have corrected the Abstract as suggested.

3) A final sentence briefly indicating the significance of the findings should be added at the end of this section.

We have added a sentence.

Results:4) the authors should include in this section additional information regarding the characterization of TMEM95 protein which could contribute to a better interpretation of the results obtained. i.e. tissue expression (testes, epididymis, accessory glands etc) and tissue specificity.

We have performed mRNA and protein analysis on different tissues demonstrating testis-specific expression of TMEM95.

5) Based on immunocytochemistry studies, the authors indicate that TMEM95 is present in the acrosome membrane of WT intact sperm. They should be aware that acrosome membrane is not the same than acrosomal region. If sperm are intact, the presence of TMEM95 in the acrosome membrane would imply this is an internal rather that a superficial protein. The authors should carry out the localization studies in permeabilized and non-permeabilized intact WT sperm in order to discriminate between the two possibilities.

As previously mentioned (essential revision 5), we have conducted IHC in permeabilizing and non-permeabilizing conditions observing TMEM95, IZUMO1 and PNA signals in non-permeabilizing conditions. Given this result, we cannot demonstrate that TMEM95 is present in the acrosome membrane. As the signal is restricted to the acrosomal cap, co-localizing with both IZUMO and PNA in intact sperm, we have modified the text using the term acrosomal cap.

Figure 1E legend indicates that "TMEM95 is localized around the acrosomal region of acrosome-intact sperm and in the sperm head after the acrosome reaction". The term "sperm head" is very vague. The authors should indicate where in the sperm head TMEM95 is localized. According to Figure 1F, TMEM95 should be located in the equatorial segment but the image in Figure 1E does not seem to show that localization. In addition to this, Figure 1E does not include any description about what the arrows are showing in each case.

We now indicate also in that figure legend that TMEM95 relocalized to the equatorial segment. Specific localization details are more difficult to spot on Figure 1E, as it is taken with a lower magnification objective, however, a wider distribution compatible with equatorial segment location can also be observed on the PNA negative sperm compared with the other PNA positive sperm on that figure.

The authors indicate that after the acrosome reaction, TMEM95 translocates to the equatorial segment. A protein can relocalize to the equatorial segment as a consequence of either its exposure due to the release of decapacitation factors or its migration from the acrosomal region. Based on the studies included in this manuscript, the authors cannot state that TMEM95 "translocates" from one region to the other.

We have changed translocation to relocalization through the manuscript.

6) In addition to the evaluation of gamete fusion by the Hoechst technique, the authors should analyze the number of sperm that bind to the egg surface. This analysis will provide information on whether or not TMEM95 participates in the first stage of gamete fusion i.e. sperm-egg binding

We have now conducted this analysis concluding that TMEM95 is not essential for sperm-egg binding.

7) Figure 3: as mentioned above, the co-immunoprecipitation assay lacks appropriate positive controls that confirm that the technique employed is in fact capable of detecting a potential interaction between TMEM95 and JUNO (i.e a control showing IZUMO/JUNO interaction). Figure 3 / Table 2: the authors carried out a series of Pull-Down experiments to identify potential TMEM95 partners in either the egg or the sperm. However, the explanation of these experiments and the interpretation of the results obtained are very poorly described.

We have conducted a new analysis using a more sensitive technique (AVEXIS) capable of detecting IZUMO1-JUNO interaction.

Discussion:The discussion about the potential molecular mechanisms underlying TMEM95 role in gamete fusion is rather confusing. The authors first suggest that TMEM95 may not require an oocyte partner. Then, they propose TMEM95 may have affinity to a yet undescribed receptor and then, based on the fact that TMEM95 acts independently from JUNO and IZUMO1 (which is not really demnstarted), they propose that TMEM95 may play an essential role rather than by single protein-to-protein interactions. The authors should more clearly conclude whether their observations support that TMEM95 participates in the first stage of gamete fusion (i.e sperm-egg binding) through a ligand-receptor mechanism or whether the protein could be acting as a fusogenic molecule capable of being intercalated into the egg lipid bilayer.

We now provide solid evidence regarding the independence of TMEM95 from the JUNO-IZUMO1 interaction (Figure 3A). As TMEM95 ablation did not disrupt sperm binding (Figure 2F), TMEM95 must be involved in post-binding events required for fertilization. To test a possible role of TMEM95 as a cell fusion promoter, we have performed a fusion assay with cells expressing TMEM95 in the presence of JUNO and IZUMO1. We have not observed cell fusion, suggesting that either TMEM95 is not involved in fusion or that, being involved, it requires the intervention of other yet unknown proteins.

[Editors' note: further revisions were suggested prior to acceptance, as described below.]

The manuscript has been significantly improved but there are some remaining issues that need to be addressed before acceptance, as outlined below. We understand that due to the current pandemic situation additional experiments might not be feasible, therefore, we ask authors to focus on the textual revisions as suggested by reviewers. Here we provide a full review as it contains many suggestions authors might find helpful.

We thank the reviewers and editors for their positive comments and appreciate the transparent reviewing process that distinguishes *eLife*. We have modified the article to fulfil the requests and have included a very recent article published in PNAS this week (Noda et al., 2020) which reports some aspects of the TMEM95 KO phenotype. We would like to stress that we were unaware of this article during the preparation, submission and revision of our current submitted article. As you will see, Our article was submitted to *eLife* one month earlier than the Noda paper was submitted to PNAS, and 5 months earlier it was submitted to PNAS, where it was editorially rejected (not sent for review) as the finding of an essential protein for gamete fusion was “not considered to have the broad appeal” required for that journal. We honestly thank *eLife* for the opportunity of publishing our results.

Reviewer #2:I am almost satisfied with the modifications provided by the authors, however, I still remain some concerns before publication in eLife.1) The authors should mention SPACA6 as a gamete fusion related factor not only in Discussion but also in Abstract and Introduction, because I think SPACA6 is an unneglectable factor for this process (Barbaux et al., 2020).

We agree. We have added Barbaux et al., 2020 and Noda et al., 2020, both of which were published during the second revision of our manuscript.

2) In Figure 2—figure supplement 1C, there is no positive control (e.g. IZUMO1) in this system.

We apologize for not using IZUMO1 as positive control for binding. The oocytes used for this study were processed as those used for sperm binding assays, proving that the procedure used (both timing after hCG injection and zona removal) did not impair sperm binding ability to the oolema.

3) I recommend a rat anti-mouse IZUMO1 monoclonal antibody clone 125 (abcam, ab211626 or BioAcademia, 73-045 or anticorps-eligne, ABIN2452040) for detection in western blotting instead of polyclonal antibody because of high reliability.

Thank you for your valuable suggestion.

Reviewer #3:[…]These studies have certainly fully answered the two main concerns of this reviewer supporting the publication of the manuscript in the journal. Nevertheless, there are some aspects that the authors should take into consideration in order to improve the quality of the manuscript.– The authors indicated that the exact localization of the protein (internal or external) could not be analyzed because the employed fixation itself produced a permeabilization of the cells. Whereas this is true for many fixation conditions, the authors should consider carrying out immunolocalization studies in live, non-fixed sperm in suspension. This is an easy approach that will allow the authors to define the protein localization in the cell. This is not a minor point as it is known that the localization of a molecule provides important information regarding the mechanisms underlying its functional role, an issue still not fully resolved for TMEM95. In this regard, the author's claim that TMEM95 likely localizes to the acrosomal membrane because both it relocalizes to the equatorial segment after the acrosome reaction and it has a TM domain are not correct as plasma membrane proteins with or without TM domains do so as well.

We acknowledge the reviewer’s concerns about TMEM95 localization, but regretfully we cannot provide more solid evidence regarding the precise localization within the acrosomal cap. Unfortunately, immunolocalization in live sperm is not an option: as TMEM95 antibody required overnight incubation, sperm would die during the incubation time in antibody solution and their membranes would become permeable. We acknowledged that TMEM95 localization within the acrosomal cap is not fully resolved and consequently we did not claim that Tmem95 localizes to the acrosomal membrane in both the previously revised version and this latest version.

– Whereas the authors have carried out the tissue expression and specificity studies requested by this reviewer, I could not find the description of the results obtained within the text but just in the figure. This should be corrected.

We have included it in the text.

– I would remove the first section within Results entitled "TMEM95 protein in silico folding suggests a role in gamete fusion" as it describes just an in silico analysis that does not really provide any new information compared to the bull protein and does not deceives an independent paragraph. I would join the results of this paragraph that describes just Figure 1A with the results of the following paragraph describing the rest of Figure 1. The authors may need to change the title of that second paragraph too.

We have now joined both sections as requested.

– The authors indicated that the lack of TMEM95 interaction with IZUMO together with the normal relocalization of IZUMO1 in TMEM95 KO sperm makes it unlikely that IZUMO1 KO would lead to the ablation of TMEM95 and that performing that experiment would delay the publication of these findings. In this regard, I think the authors could contemplate the possibility of sending their anti-TMEM95 antibody to Dr Ikawa who could perform this easy localization study in his lab. Of note in this regard, the authors already have contact with Dr Ikawa's lab which has provided the anti-IZUMO antibody to them.

During this second revision of this manuscript, Ikawa´s group has published an article reporting roles of TMEM95, SPACA6 and SOF1 in fertilization (Noda et al., 2020). In that article they found that the relocalization of IZUMO1 was not impaired by the ablation of any of these three genes (in agreement with our results and Barbaux et al., 2020). Unfortunately, they did not test the opposite (i.e., whether IZUMO1 ablation impaired the other proteins), as sperm localization was only shown for SPACA6 and this analysis was not performed in *Izumo1* KO sperm. As the antibody we used in this article is commercially available, we believe that they will be able to perform this experiment in subsequent publications.

– In my opinion, there are several observations which are not sufficiently discussed. The authors may speculate about several issues such as the finding that TMEM95 and IZUMO are both necessary but not sufficient for gamete fusion, how the protein reaches the equatorial segment, potential ways to identify TMEM95 partners, whether TMEM95 is present or not in IZUMO KO cells, among others, which will certainly enrich the discussion.

We have now extended the Discussion while avoiding to be too speculative. For instance, we cannot say much about TMEM95 relocalization, as the precise location within the acrosomal cap could not be determined.